



# 1 Measurement Report: Distinct size dependence and

# 2 diurnal variation of OA hygroscopicity, volatility, and

# 3 CCN activity at a rural site in the Pearl River Delta

# 4 (PRD) region, China

Mingfu Cai[1,2,3], Shan Huang[1,2*], Baoling Liang[4], Qibin Sun[4], Li Liu[5*], Bin Yuan[1,2],
Min Shao[1,2], Weiwei Hu[6], Wei Chen[6], Qicong Song[1,2], Wei Li[1,2], Yuwen Peng[1,2],
Zelong Wang[1,2], Duohong Chen[7], Haobo Tan[5], Hanbin Xu[4], Fei Li[5], Xuejiao Deng[5],
Tao Deng[5], Jiaren Sun[3], and Jun Zhao[4,8,9]
[1] Institute for Environmental and Climate Research, Jinan University, Guangzhou, Guangdong
511443, China
[2] Guangdong-Hongkong-Macau Joint Laboratory of Collaborative Innovation for Environmental
Quality, Guangzhou, Guangdong 511443, China
[3] Guangdong Province Engineering Laboratory for Air Pollution Control, Guangdong Provincial
Key Laboratory of Water and Air Pollution Control, South China Institute of Environmental
Sciences, MEE, Guangzhou, Guangdong 510655, China
[4] School of Atmospheric Sciences, Guangdong Province Key Laboratory for Climate Change and
Natural Disaster Studies, and Institute of Earth Climate and Environment System, Sun Yat-sen
University, Zhuhai, Guangdong 519082, China
[5] Institute of Tropical and Marine Meteorology of China Meteorological Administration, Guangzhou
510640, China
[6] State Key Laboratory of Organic Geochemistry and Guangdong Key Laboratory of Environmental
Protection and Resources Utilization, Guangzhou Institute of Geochemistry, Chinese Academy of
Sciences, Guangzhou 510640, China
[7] Guangdong Environmental Monitoring Center, Guangzhou 510308, China
[8] Southern Marine Science and Engineering Guangdong Laboratory (Zhuhai), Zhuhai, Guangdong



519082, China
[9] Guangdong Provincial Observation and Research Station for Climate Environment and Air Quality
Change in the Pearl River Estuary, Guangzhou, Guangdong 510275, China
*Corresponding authors: Shan Huang (shanhuang_eci@jnu.edu.cn) and Li Liu (liul@gd121.cn)



**Abstract.**
Organic aerosol (OA) has a significant contribution to cloud formation and hence climate
change. However, high uncertainties still exist in its impact on global climate, owing to the varying
physical properties affected by the complex formation and aging processes. In this study, the
hygroscopicity, volatility, cloud condensation nuclei (CCN) activity, and chemical composition of
particles were measured using a series of online instruments at a rural site in the Pearl River Delta
(PRD) region of China in Fall 2019.  During the campaign, the average hygroscopicity of OA ($\kappa_{OA}$)
increased from 0.058 at 30 nm to 0.09 at 200 nm, suggesting a higher oxidation state of OA at larger
particle sizes, supported by a higher fraction of extremely low volatile OA (ELVOA) for larger size
particles. Significantly different diurnal patterns of $\kappa_{OA}$ were observed between Aitken mode and
accumulation mode. For Aitken mode (30-100 nm), the $\kappa_{OA}$ values showed daily minima (0.02-0.07)
during daytime, while exhibited a daytime peak (~0.09) in the accumulation mode. Coincidently, a
daytime peak was observed for both aged biomass burning organic aerosol (aBBOA) and less
oxygenated organic aerosol (LOOA) based on source apportionment, which were attributed to the
aging processes and gas-particle partitioning through photochemical reactions. In addition, the
fraction of semi-volatile OA (SVOA) was higher at all measured sizes during daytime than during
nighttime. These results indicate that the formation of secondary OA (SOA) through gas-particle
partitioning can generally occur at all diameters, while the aging processes of pre-existing particles
are more dominated in the accumulation mode. Furthermore, we found that applying a fixed $\kappa_{OA}$
value (0.1) could lead to an overestimation of the CCN number concentration ($N_{CCN}$) up to 12%-
19% at 0.1%-0.7% supersaturation (SS), which was more obvious at higher SS during daytime.
Better prediction of $N_{CCN}$ could be achieved by using size-resolved diurnal $\kappa_{OA}$, which indicates that



the size-dependence and diurnal variations of $\kappa_{OA}$ can strongly affect the $N_{CCN}$ at different SS. Our
results highlight the need for accurately evaluating the atmospheric evolution of OA at different size
ranges, and their impact on the physicochemical properties and hence climate effects.

## 56    1.    Introduction

The impact of aerosol particles on global climate is widely known, including absorbing and

scattering solar radiation, and acting as cloud condensation nuclei (CCN). However, the extent of
their contribution on the climate forcing is still unclear. Organic aerosol (OA) as a dominant
component of fine particles (Jimenez et al., 2009), may contribute the uncertainties of climate effects
of particles, mainly owing to unknown sources, physical properties, formation, and aging
mechanisms (Volkamer et al., 2006;Kuang et al., 2020b;Rastak et al., 2017). Numerous studies show
that secondary organic aerosol (SOA) accounts for a large OA fraction in most atmospheric
environments (e.g., Huang et al., 2014;Shrivastava et al., 2017;Kanakidou et al., 2005;Hallquist et
al., 2009). Nevertheless, both primary OA (POA) and SOA in the ambient air remain poorly
characterized in terms of the formation mechanism and atmospheric evolution, and their particle
diameter can vary on a large scale. Their impact on the global climate and atmospheric chemistry is
still highly uncertain.

A combination of Aerodyne Aerosol Mass Spectrometer (AMS) or Aerosol Chemical Species

Monitor (ACSM) with positive matrix factorization (PMF) is widely used for investigating the OA
evolution in the atmosphere (Li et al., 2013;Huang et al., 2018;Huang et al., 2014;Chen et al.,


2015;Jimenez et al., 2009). For instance, Qin et al. (2017) found that hydrocarbon-like OA (HOA)
from traffic emission contributed up to 40% of OA during nighttime, owing to daytime traffic
restrictions on heavy vehicles in urban Guangzhou. Kuang et al. (2020a) reported a dominant
contribution to oxygenated OA (OOA) through aqueous-phase reaction in the North China Plain
(NCP). Guo et al. (2020) observed different SOA mechanisms between clean and pollution episodes
in the Pearl River Delta (PRD) region. Nevertheless, the investigation of bulk OA is still insufficient
in understanding the aerosol climate effects without the size-resolved characterization. The OA size
distribution is largely dependent on its composition, sources and aging level. Li et al. (2012)
observed various mass distribution patterns for different species in airborne particulate organics and
reported that dehydrated sugars, fossil fuel-derived *n*-alkanes, and PAHs showed a unimodal
distribution, while non-dehydrated sugars and plant was derived as n-alkanes which presented a
bimodal pattern. In the urban region, Aitken mode was mainly dominated by HOA owing to traffic
emissions (Zhang et al., 2005b;Cai et al., 2020). In the marine atmosphere, the size distribution of
fine mode POA was found to be independent of sea salt, while coarse mode particles tended to be
internally mixed with sea salt (Gantt and Meskhidze, 2013). Similarly, the OA physical properties
were also found to be size-dependent. Deng et al. (2018) reported a higher OA hygroscopicity
($\kappa_{OA} \approx 0.22$) at about 150 nm than that ($\kappa_{OA} \approx 0.19$) at sub-100 nm. In contrast, Zhao et al. (2015)
found that $\kappa_{OA}$ decreased from 0.17 at 50 nm to 0.07 at 200 nm in laboratory study, which was
attributed to the higher oxidation degree for smaller particles.

The size dependence of OA properties in the aforementioned studies might exert impact on the

CCN prediction, which is mainly determined by their sources and formation processes. Cai et al.
(2018) found that $N_{CCN}$ at 0.1% SS was underestimated by about 10% if a $\kappa_{OA}$ value of 0.1 was used.



94 A model simulation from Liu and Wang (2010) showed that an increase of about 40-80% for the

95 CCN concentration was obtained by increasing the κ value of POA from 0 to 0.1. Wang et al. (2008)

96 reported that the uncertainties of the first indirect aerosol effect varied from -0.2 to 0.2 W m$^{-2}$ for a

97 $\kappa_{OA}$ value of 0 to 0.25. Rastak et al. (2017) showed that using a single-parameter framework of $\kappa_{OA}$

98 in evaluating the climate effects of aerosol could lead to significant errors (about -1.02 W m$^{-2}$),

99 which is the same order as the climate forcing of anthropogenic aerosol during the industrial period.

100 These results further highlight a need for the understanding of the relationship between the OA

101 evolution processes and its impact on the CCN activity at different particle sizes.

102  The OA hygroscopicity and volatility can provide information about the evolution of OA, given

103 that they are often related to the chemical composition of the particles. A positive correlation

104 between the hygroscopicity values and the oxidation degree of OA, including the ratio of atomic

105 oxygen to atomic carbon (O:C), the oxidation state ($\overline{OS_C}$), or the mass fraction of m/z 44 (for $CO_2^+$)

106 ion fragments in the organic spectra ($f_{44}$) from chemical composition, were widely reported in the

107 literature (Wu et al., 2013;Pajunoja et al., 2015;Chang et al., 2010). Kim et al. (2020) found that the

108 $\kappa_{OA}$ was positively and negatively correlated with OOA and HOA at different size ranges,

109 respectively. Deng et al. (2019) reported a decreasing trend of $\kappa_{OA}$ at a size range of 100-360 nm

110 during daytime in a forest environment, suggesting the formation of biogenic SOA (BSOA) through

111 photochemical oxidation of biogenic volatile organic compounds (BVOCs). The OA volatility,

112 specifically saturation vapor concentration ($C^*$), is linked to the gas-particle partitioning and aging

113 processes. In general, the $C^*$ value decreases with an increase of the oxidation degree and the

114 number of atomic carbon (Donahue et al., 2011). May et al. (2013) found that most of the biomass-

115 burning POA were semi-volatile. Saha et al. (2017) showed a lower volatility of OA in the afternoon



hours using a dual-thermodenuder (TD) system, probably owing to photochemical oxidation of OA.
Hong et al. (2017) derived the OA volatility distribution by a combination of the VTMDA
measurement and a multi-component evaporation dynamics model, and found a moderate (R≈0.4)
correlation between the OA groups obtained by the VTDMA data and the PMF results, respectively.

In this study, we investigate physical properties of OA at different size ranges, and evaluate

their influence on the atmospheric CCN concentration. A rural field measurement was conducted at
the Heshan site in the Pearl River Delta (PRD) region, China, during Fall 2019 (October and
November). The hygroscopicity, volatility, size-resolved CCN activity, and chemical composition
were measured by a series of online instruments. The size-resolved hygroscopicity and volatility
distribution of organics was investigated. PMF was employed to analyze the sources and processes
of OA. The impact of diurnal variation and size dependence of $\kappa_{OA}$ on the $N_{CCN}$ prediction at
different supersaturation (SS) was assessed.
**2.    Measurement and methodology**
**2.1 Measurement site**

The field measurements were conducted at the Heshan supersite in the Guangdong province,

China during autumntime 2019 (27[th] September to 17[th] November 2019). This supersite (22°42′39.
1″N, 112°55′35.9″E) is located at the southwest of the PRD region and surrounded by farms and
villages, with an altitude of about 40 m. All sample particles first passed through a Nafion dryer
(Model MD-700, Perma Pure Inc., USA) to maintain a relative humidity (RH) lower than 30%. The
schematic diagram of the experimental setup can be found in Fig. S1. Detailed descriptions of the



measurement site and some instruments can be found in Cai et al. (2021a).

## 2.2 Instrumentation

### 2.2.1    Aerosol hygroscopicity and volatility measurement

Size-resolved hygroscopicity and volatility of particles were measured by a H/V-TDMA
(model M3000, Bmet Inc., China). The instrument consists of two differential mobility analyzers
(DMA1 and DMA2, model 3081 L, TSI Inc., USA), a Nafion humidifier (Model MD-700, Perma
Pure Inc., USA), a heater tube, and a condensation particle counter (CPC, model 3788, TSI Inc.,
USA). The instrument was operated in H- and V- mode during the measurement with a cycle time
of about 3-4 h. The dried sample particles were firstly charged by an X-ray neutralizer and then
classified by DMA1 at six diameters (30, 50, 80, 100, 150, and 200 nm). In the H-mode, the chosen
particles with a specific dry diameter ($D_0$) were sequentially humidified by the Nafion humidifier
to achieve 90% of RH. A combination of DMA2 and CPC were employed to measure the size
distribution of humidified particles ($Dp_{wet}$). The hygroscopic growth factor (GF) at a certain dry
diameter can be defined as:
$$GF(D_0) = \frac{Dp_{wet}}{D_0} \tag{1}$$
In the V-mode, the selected particles from DMA1 were heated in the heater tube at 100, 150,
200, and 250°C, respectively. Similar to the H mode, the size distribution of heated particles along
with particles at room temperature (25°C) was measured by the DMA2 and CPC. The volatility
shrink factor (VSF) at a certain diameter and temperature is then defined as:
$$VSF(T, D_0) = \frac{Dp(T)}{D_0} \tag{2}$$
Before the campaign, standard polystyrene latex spheres (PSLs; with a size of 20, 50, and 200



nm), ammonium sulfate, and sodium chloride were used to calibrate the diameter classification of
DMAs, hygroscopicity measurement, and the transport efficiency of particles in the heater tube,
respectively. For the H/V-TDMA data, the TDMAfit algorithm (Stolzenburg and McMurry, 2008)
was applied to fit the growth factor and volatility shrink factor probability density function (GF-
PDF and VSF-PDF) with various DMA transfer functions. The detailed data inversion processes
can be found in Tan et al. (2013a).
**2.2.2      The size-resolved CCN activity and particle number size distribution measurement**

A combination of a cloud condensation nuclei counter (CCNc, model 200, DMT Inc., USA)

and a scanning mobility particle sizer (SMPS, model 3938L75, TSI Inc., USA) was employed to
measure size resolved CCN activity. The supersaturation (SS) of each column (A and B) of CCNc
was set to be 0.1%, 0.2% and 0.4% (for column A), and 0.7%, 0.9% and 1.0% (for column B),
respectively. During the measurement, the SMPS was operated at a scanning mode. The sample
particles were firstly neutralized by an X-ray neutralizer (model 3088, TSI Inc., USA) and were
subsequently classified by a DMA. The classified particles were then split into three paths: one to a
CPC (model 3756, TSI Inc., USA) for measurement of particle number concentration (with a flow
rate of 0.6 LPM) and two to the CCNc for measurement of the CCN number concentration ($N_{CCN}$)
at a specific SS (with a flow rate of 0.5 LPM).

The particle number size distribution in a size range of 1 nm-10 μm was measured by a suite of

instruments including a diethylene glycol scanning mobility particle sizer (DEG-SMPS, model
3938E77, TSI Inc., USA), a SMPS (model 3938L75, TSI Inc., USA), and an aerodynamic particle
sizer (APS, model 3321, TSI Inc., USA). The detailed description of these instruments can be found



in Cai et al. (2021a). Before the measurement, the SMPSs were calibrated with PSLs (20, 50 and
200 nm) and the CCNc was calibrated with ammonium sulfate ($(NH_4)_2SO_4$) particles at selected SSs
(0.1%, 0.2%, 0.4%, 0.7%, 0.9%, and 1.0%).
**2.2.3    Aerosol chemical composition measurement**

The size-resolved chemical composition of ambient aerosol particles was measured by a soot

particle aerosol mass spectrometer (SP-AMS, Aerodyne Research, Inc., USA). The principle and
operation of the instrument are generally the same as a high resolution time-of-flight aerosol mass
spectrometer (HR-ToF-AMS) (Canagaratna et al., 2007). In addition to an original tungsten
vaporizer (~600°C), a soot-particle module which mainly contains a Nd:YAG (1064 nm) laser was
integrated into HR-ToF-AMS for vaporizing refractory species (Onasch et al., 2012). As a result,
SP-AMS can provide chemical information for non-refractory species (nitrate, sulfate, ammonium,
chloride, and organics) as well as refractory species such as refractory black carbon (rBC) and
several metals. During the campaign, SP-AMS was run between V mode (only tungsten vaporizer)
and SP mode (tungsten and laser vaporizers) with a time resolution of 1 min. In order to minimize
disturbance caused by mode switch, 15 min averaged data are used in the present study. More details
on the quantification using ionization efficiency, composition dependent collection efficiency and
external instrument as well as software for SP-AMS data analysis could be found in Kuang et al.

(2021).

Facilitated by the time-of-flight chamber in SP-AMS, the particle mass size distribution can be

measured in submicrometer size range, specifically, 40 to 800 nm in vacuum aerodynamic diameter
(Dva). The mass size distribution for relevant AMS species was used in this study for investigating



the link between particle chemical composition and volatility/hygroscopicity. Since SP-AMS
provided the size distribution versus Dva, the equation below was used to convert Dva into mobility
diameter (Dp).
$$D_p = \frac{D_{va}}{S \times \frac{\rho_p}{\rho_0}}$$    (3)
where S is the shape factor, $\rho_p$ is the particle density, and $\rho_0$ is the density for water (1 kg m$^{-3}$). In
this study, we estimate that the particles were close to sphere due to high RH in the PRD and thus a
shape factor of 0.8 was applied. An overall particle density of 1.6 kg m$^{-3}$ is used.

Based on high resolution data from SP-AMS, source apportionment was performed for organic

aerosols (OA) in the bulk PM$_1$ with positive matrix factorization (PMF, Paatero, 1997;Paatero and
Tapper, 1994) following the instruction in Ulbrich et al., 2009. The input data, selection of solutions,
mass spectral profile, and time series of each factor can be found in Kuang et al. (2021). In brief,
OA measured at the Heshan site could be divided into six components with identified sources and
processes, including two from primary sources and four factors corresponding to secondary
formation: a hydrocarbon-like OA (HOA) contributed mainly by vehicle exhausts mixed with
cooking emissions, a biomass burning OA (BBOA) related to biomass burning combustion from the
surrounding villages, an aged BBOA (aBBOA), a more oxygenated OA (MOOA) from regional
transport, a less oxygenated OA (LOOA) provided by daytime photochemical formation, and a
nighttime-formed OA (Night-OA) related to secondary formation during nighttime.
**2.3 Methodology**
**2.3.1    Estimates of hygroscopicity**

The hygroscopicity parameter κ can be obtained under subsaturation condition by the H/V-





TDMA measurement and supersaturation condition by the CCNc measurement. The $\kappa$ value
($\kappa_{HTDMA}$) can be estimated from the growth factor measured by H/V-TDMA (Petters and
Kreidenweis, 2007):
$$\kappa_{HTDMA} = (GF^3 - 1)\left[\frac{1}{RH}\exp\left(\frac{4\sigma_{s/a}M_w}{RT\rho_w D_p} - 1\right)\right] \tag{4}$$
where $\sigma_{s/a}$ is the surface tension of the solution/air interface and the solution is temporarily assumed
to be pure water (0.0728 N m$^{-1}$ at 298.15 K), $M_w$ is the molecular weight of water (0.018 kg mol$^{-1}$),
R is the universal gas constant (8.31 J mol$^{-1}$ K$^{-1}$), T is the thermodynamic temperature in Kelvin
(298.15 K), $\rho_w$ is the density of water (about 997.04 kg m$^{-3}$ at 298.15 K) and $D_p$ is the particle
diameter in meter.
For the CCNc measurement, the $\kappa$ value ($\kappa_{CCN}$) is calculated from the critical supersaturation
(Sc) and the critical diameter ($D_{50}$) by the following equation (Petters and Kreidenweis, 2007):
$$\kappa_{CCN} = \frac{4A^3}{27D_{50}^3(\ln Sc)^2}, A = \frac{4\sigma_{s/a}M_w}{RT\rho_w} \tag{5}$$
The critical diameter, $D_{50}$, is defined as the diameter at which 50% of the particles are activated
at a specific SS, and can be obtained from the $N_{CCN}$ and $N_{CN}$ measured by the CCNc and SMPS
system:
$$\frac{N_{CCN}}{N_{CN}} = \frac{B}{1+(\frac{D_p}{D_{50}})^c} \tag{6}$$
where the B and C are fitting coefficients.
**2.3.2    Derivation of the size-resolved hygroscopicity of organic matter**
The size-resolved chemical composition is adopted to derive the size-dependent hygroscopicity
of organic matter ($\kappa_{OA}$). However, the AMS cannot provide sufficient information of the size-
resolved species, especially for small size particles (< 100 nm) owing to the low mass concentration.



Thalman et al. (2017) proposed a method to reconstruct the size-resolved chemical composition,
which combines a time-resolved bulk mass concentration and an average mass distribution.
Nevertheless, the variation of mass distribution was not considered in this method. In this study, a
bimodal lognormal distribution function method was adopted and the one-hour average mass
distribution was fitted to obtain the reconstructed size-resolved chemical composition. The average
mass distribution with bimodal lognormal fitted modes of each species was shown in Fig. S2.

According to the ZSR mixing rule (Zdanovskii, 1948;Stokes and Robinson, 1966), the

hygroscopicity of particles ($\kappa_{AMS}$) can be calculated based on the SP-AMS measurement, assuming
an internal mixing state for all particles:

$$\kappa_{AMS} = \sum_i \kappa_i \varepsilon_i \tag{7}$$

where $\kappa_i$ is the $\kappa$ value of each component and $\varepsilon_i$ is the volume fraction of corresponding species in
particles. The mole concentrations of the inorganic species are estimated based on the $NH_4^+$, $SO_4^{2-}$,
and $NO_3^-$ measured by the AMS (Gysel et al., 2007):

$$n_{NH_4NO_3} = n_{NO_3^-}$$

$$n_{H_2SO_4} = max\,(0, N_{SO_4^{2-}} - n_{NH_4^+} + n_{NO_3^-})$$

$$n_{NH_4HSO_4} = min\left(2n_{SO_4^{2-}} - n_{NH_4^+} + n_{NO_3^-}, n_{NH_4^+} - n_{NO_3^-}\right)$$

$$n_{(NH_4)_2SO_4} = max\left(n_{NH_4^+} - n_{NO_3^-} - n_{SO_4^{2-}}, 0\right)$$

$$n_{HNO_3} = 0 \tag{8}$$

where n denotes the number of moles of each component ($NH_4^+$, $SO_4^{2-}$ and $NO_3^-$), $\varepsilon_{org}$ and $\varepsilon_{BC}$
were obtained from mass concentration measured by the SP-AMS. The density and $\kappa$ value of each
component were listed in Table 1.

The $\kappa_{OA}$ can be calculated based on the size-resolved chemical composition and H/V-TMDA



measurement using following equation:
$$\kappa_{OA} = \frac{\kappa_{HTDMA} - (\kappa_{inorgsalt}\varepsilon_{inorgsalt} + \kappa_{BC}\varepsilon_{BC})}{\varepsilon_{org}}$$ (9)
**2.3.3**     **Volatility data**
During the heating process, some particles could be lost between DMA$_1$ and DMA$_2$ due to
complete evaporation (CV), thermophoresis, and Brownnian diffusion (Philippin et al., 2004).
Owing to these losses, the V-mode measurement does not represent the actual volatility distribution
of the monodisperse particles. The sodium chloride (NaCl) particles, which do not evaporate at the
set temperature in this measurement, were used to determine the particle losses owing to
thermophoretic forces and diffusion. The size- and temperature-dependent transmission efficiency
($\eta(D_p, T)$) of NaCl in the heater was shown in Fig. S3. Thus, the number fraction of CV group
($NF_{CV}(D_p, T)$) at a certain diameter and temperature can be calculated using the following equation
(Cheung et al., 2016):
$$NF_{CV}(D_p, T) = 1 - \frac{N'(D_p, T)}{N(D_p)\eta(D_p, T)}$$ (10)
where $N'(D_p, T)$ is the number concentration of particles at a specific diameter and temperature
after heating, which was measured by the CPC in the H/V-TDMA. The $N(D_p)$ is the number
concentration of particles with a diameter $D_p$ before heating, which was provided by the SMPS
measurement. The volume fraction remaining (VFR) after heating for the measured particles can be
obtained according to the following equation:
$$VFR(D_p, T) = \sum_i VSF_i^3(D_p, T) NF_i(D_p, T)[1 - NF_{CV}(D_p, T)]$$ (11)
where $i$ represents the $i$th VSF bin, and $NF_i$ is the number fraction of particles with $VSF_i$, which is
calculated based on the VSF-PDF ($c(VSF, D_p, T)$):



$NF_i = \int_{VSF_i}^{VSF_{i+1}} c(VSF, D_p, T)\, dVSF$      (12)
The mass fraction remaining (MFR) was assumed to be proportional to VFR, assuming that
the density of particles was constant before and after heating.
**2.3.4      Multi-component evaporation dynamics model**
Based on the volatility basis set (VBS) framework (Donahue et al., 2011), the organic matter
was classified into three organic groups based on the saturation concentration (C*($T_{ref}$),
$T_{ref}$=298.15 K): extremely low volatility organic aerosol (ELVOA, $C^*$=10$^{-5}$ µg m$^{-3}$), low volatility
organic aerosol (LVOA, $C^*$=10$^{-2}$ µg m$^{-3}$), and semi-volatility organic aerosol (SVOA, $C^*$=10 µg m$^-$
$^3$).
A multi-component evaporation dynamics model described by Lee et al. (2011) was used to
simulate the evaporation of particles in the heated tube of the H/V-TDMA by solving the mass
transfer regime equation, in order to obtain the size-resolved distribution of the aforementioned
three OA groups. The MFR, residence time (about 4.11 s) in the heater tube, the temperature of the
heater tube, particle number concentration, particle sizes, chemical composition, and
thermophysical properties of each species (Table 2) were input into the model. The particles were
assumed to be internally mixed with organic and inorganic species, including three organic groups,
NH$_4$NO$_3$, (NH$_4$)$_2$SO$_4$, and black carbon (BC). The mass transfer of each component $i$ between the
aerosol and gas phases in the transition regime was calculated from the following equation:
$\frac{dm_{p,i}}{dt} = 2\pi D_i D_p f(Kn, \alpha)\left(C_{i,g} - f_i C_i^*(T) exp\left(\frac{4\sigma_{s/a} M_i}{D_p \rho_i RT}\right)\right)$
$\frac{dC_{i,g}}{dt} = -\frac{dm_{p,i}}{dt} N_p(D_p)$      (13)
where $m_{p,i}$ (µg) is the mass of species $i$ in a single particle, $C_{i,g}$ (µg m$^{-3}$) is its gas-phase





concentration, $D_i$ (m$^2$ s$^{-1}$) is the diffusion coefficient for species $i$ in air, $D_p$ (m) is the particle
diameter, $f(Kn, \alpha)$ is a correction term to account for non-continuum mass transfer depending on
Knudsen number ($Kn$) and mass accommodation coefficient ($\alpha$), $f_i$ is the mole fraction of species
$i$, $C_i^*(T)$ is the saturation concentration at temperature ($T$) of the heater tube, $M_i$ (kg mol$^{-1}$) is the
molecular weight of species $i$, $\rho_i$ (kg m$^{-3}$) is its density and $N_p(D_p)$ (cm$^{-3}$) is the number
concentration of particles with a diameter $D_p$.
The correction term $f(Kn, \alpha)$ is determined by the following equation (Seinfeld and Pandis,

2016):

$f(Kn, \alpha) = \frac{1+Kn}{1+2Kn(1+Kn)/\alpha}$
$Kn = \frac{2\lambda_i}{D_p}$     (14)
where $\lambda_i$ is the mean free path of species $i$ in the air, which is defined as $\lambda_i = \frac{2D_i}{c_i}$. The $c_i$ is the mean
speed of species $i$ and $c_i = \sqrt{\frac{8RT}{\pi M_i}}$.
The temperature-dependent $C_i^*(T)$ is estimated from the Clausius-Clapeyron equation:
$C_i^*(T) = C_i^*(T_{ref})exp\left[\frac{\Delta H_{vap,i}}{R}\left(\frac{1}{T_{ref}} - \frac{1}{T}\right)\right]\frac{T_{ref}}{T}$     (15)
where $\Delta H_{vap,i}$ (kJ mol$^{-1}$) is the enthalpy of vaporization. The known mass fractions of NH$_4$NO$_3$,
(NH$_4$)$_2$SO$_4$, and BC were calculated respectively, based on the SP-AMS measurement. The time
step of the model was set to be 10$^{-3}$ s. The characteristics of each species were listed in Table 2. The
mass fraction of each organic group in different particle sizes was derived by minimizing the squared
residuals (SSR) values, $SSR = \sum_{T_i=T_1}^{T_5}[MFR_{model}(T_i, Dp) - MFR_{measured}(T_i, Dp)]^2$. The non-
linear constrained optimization function "fmincon" in MATLAB (version 2016a, Mathworks Inc.)
was used to obtain the optimal fitted result. A constrained of $\sum f_{i,inorganics} + \sum f_{i,organics} = 1$ is used.
The modeled MFR is strongly dependent on the values of vaporization enthalpy ($\Delta H_{vap}$) and





mass accommodation coefficient ($\alpha$) (Lee et al., 2010;Lee et al., 2011). Thus, a sensitivity test is
performed to determine the $\Delta H_{vap}$ of OA and $\alpha$ based on the campaign average data (Fig. S4). A
linear relationship was adopted between $\Delta H_{vap}$ and $log_{10} C_i^*(T_{ref})$ , $\Delta H_{vap} = -a \cdot$
$log_{10} C_i^*(T_{ref}) + b$, where a and b are fitting parameters (Epstein et al., 2010). The a and b values
are set to be [0, 4, 8, 12] and [50, 80, 100, 150, 200] in the sensitivity test, respectively, along with
$\alpha$ = [0.01, 0.09, 0.1, 0.5, 0.7, 0.9, 1]. The results show that the measured MFR was reproduced well
(with the lowest SSR of 0.0205, Fig. S5) by using $\Delta H_{vap}$=80 kJ mol$^{-1}$ with $\alpha$ of 0.09, 0.1 and 0.7,
respectively. For simplicity, $\Delta H_{vap}$=80 kJ mol$^{-1}$ and $\alpha$=0.09 are considered as the best estimation
and adopted in the simulation of the whole campaign datasets. The extracted $\alpha$ value was consistent
with the values ($\alpha \leq 0.1$) reported previously (Saha et al., 2015;Park et al., 2013;Saleh et al.,
2008;Cappa and Jimenez, 2010), indicating significant resistance to mass transfer during
evaporation. In addition, the $\Delta H_{vap}$ of OA is of the same magnitude (80-150 kJ mol$^{-1}$) as reported
in the literature (Hong et al., 2017;Saha et al., 2017;Riipinen et al., 2010).

Note that the decomposition of particles during the heating process is ignored in the model.

Kiyoura and Urano (1970) suggested that ammonium sulfate would decompose to ammonium
bisulfate (NH$_4$HSO$_4$) or triammonium hydrogen sulfate (NH$_4$)$_3$H(SO$_4$)$_2$, and ammonia (NH$_3$) when
heated to around 160-180 °C. Wang and Hildebrandt Ruiz (2018) also observed thermal
decomposition of organics and ammonium sulfate during evaporation by using a Filter Inlet for
Gases and AEROsols chemical-ionization mass spectrometer (FIGAERO-CIMS). It suggests that,
besides sublimation, decomposition might occur during evaporation of particles. However, the
mechanisms of decomposition are complex and remain unclear, which is difficult to simulate in our
model. We hence exclude the decomposition of particles from the model for simplicity.



## 3  Results and discussion

### 3.1  Overview

Figure 1 shows the temporal profile of PNSD (a), aerosol chemical composition and total mass

concentration of $PM_{2.5}$ (b), mass fraction of each component (c), and wind speed and direction (d)

during the measurements. Note that the SP-AMS measurement started on 12th October. According

to the PNSD data, a total number of 20 new particle formation (NPF) events were observed during

the whole campaign. The background particles mainly exhibited unimodal distribution which

peaked at a size range of about 80-150 nm. The average particle number concentration ($N_{CN}$) in the

size range of 3-1000 nm was about 12700 $cm^{-3}$, much lower than that from the rural measurement

(18150 $cm^{-3}$) in 2006 in the PRD region (Rose et al., 2010). A wide accumulation mode was

observed during the period prevalent with north wind direction, implying that the air mass from the

north could bring pollutants from the city cluster around Guangzhou to the measurement site.

The chemical composition and the corresponding mass fraction measured by the SP-AMS (Fig.

1 b and c) were consistent with those of PNSD, which showed a significantly high mass

concentration of organics when the wind was from the north. The average mass fraction of $PM_1$ was

dominated by organics (51.8%), followed by sulfate (17.5%), nitrate (10.2%), BC (9.9%),

ammonium (8.8%), and chloride (1.7%). The mass concentration of organics varied from 3.3 to

123.4 $\mu g\ m^{-3}$, with an average value of 20.3 $\mu g\ m^{-3}$, lower than the value (25.7 $\mu g\ m^{-3}$) reported in

Guangzhou city (Qin et al., 2017), but significantly higher than that was observed (4.1 $\mu g\ m^{-3}$) in

Hongkong (Lee et al., 2013). The mass distribution of the chemical species at the Heshan site was

similar to that measured in inland China (Chen et al., 2015;Huang et al., 2014), which was





dominated by organics from anthropogenic emissions. A distinguished and reproducible diurnal
pattern of the mass fraction was observed during the measurement (Fig. 1c), implying that the
particle composition was more affected by local emission or photochemical production than other
pathways. Organics showed a diurnal pattern with bimodal peaks respectively in the afternoon and
evening, which will be discussed later in section 3.3.   The temporal profile of GF-PDF (Fig. 2)
measured by the H/V-TDMA was consistent with that of chemical composition, which showed a
significant diurnal pattern. It suggested that particles at all diameters could be affected by
atmospheric chemical processes and local emissions, which will be further discussed in section 3.3.
The H/V-TDMA data from 18$^{th}$ to 26$^{th}$ October and 29$^{th}$ October to 3$^{rd}$ November were not available
due to instrumental failure. In general, the GF-PDF exhibited a bimodal distribution for particles
larger than 30 nm, with a significant more-hygroscopic (MH, GF>1.33) or less-hygroscopic (LH,
1.11<GF<1.33) mode and a less obvious non-hygroscopic (NH, GF<1.11), indicating that these
particles were partly externally mixed. The NH mode with primary emissions (e.g., fresh black
carbon and some organics) was more obvious in a size range of 50-150 nm than others, suggesting
that these particles were more affected by local anthropogenic emissions. The above observation
was supported by the size distribution of the BC mass faction (Fig. S6), which peaked at a size range
of about 50-150 nm. Besides, the MH mode shifted to a higher GF value with an increase of particle
sizes, implying that larger particles were more aged with a higher fraction of inorganic salt (Fig. S6)
and well separated from the freshly emitted counterparts. A similar phenomenal pattern was
previously observed in the urban environment, including the PRD region (Hong et al., 2018;Cai et
al., 2017;Jiang et al., 2016;Tan et al., 2013b), the North China Plain (Liu et al., 2011;Ma et al., 2016)
and other city regions around the world (Yuan et al., 2020;Mochida et al., 2006;Massling et al.,



2005).

Table 3 summaries the $N_{CCN}$, activation ratio (AR), $D_{50}$, and $\kappa_{CCN}$ values at 0.1%, 0.2%, 0.4%,
0.7%, 0.9%, and 1.0% SS during the campaign. The activation ratio is defined as the ratio of $N_{CCN}$
to $N_{CN}$, that is, $AR = N_{CCN}/N_{CN}$. The average $N_{CCN}$ at 0.1%, 0.2%, 0.4%, 0.7%, 0.9%, and 1.0% SS
was about 2507, 4322, 5854, 6834, 7497, and 7862 cm$^{-3}$, respectively. The $N_{CCN}$ at 0.7% SS was
lower than that measured (7900 cm$^{-3}$ at 0.7% SS) in urban Guangzhou (Cai et al., 2018) and at a
suburban site (14400 cm$^{-3}$ at 0.864% SS) in the North China Plain (Zhang et al., 2020), but
significantly higher than that measured at an urban site (2776 cm$^{-3}$ at 0.68% SS) in São Paulo, Brazil
(Almeida et al., 2014). The average AR at the above six SS was 0.20, 0.34, 0.45, 0.52, 0.57, and
0.60, respectively. The AR at 0.7% SS was lower than the measured value (0.64 at 0.7% SS) in the
urban Guangzhou (Cai et al., 2018), while the corresponding $D_{50}$ (52.56 nm) was lower than that
(58.45 nm) in the Guangzhou campaign, implying a higher CCN activity at this site. Thus, the lower
AR in this autumn campaign suggested that particles were more centered at smaller sizes, which
might be attributed to frequently occurred NPF at the Heshan site. The hygroscopicity parameter κ
obtained by the CCNc method were 0.48, 0.47, 0.31, 0.22, 0.20, and 0.20 at the above SS,
respectively, which was much higher than those measured by the HTDMA in this study. The
average κ values obtained using HTDMA fall in a range of 0.1-0.17 at 30-200 nm (Fig. S7), which
were possibly attributed to high fraction of organic matter (Fig. S6). This significant discrepancy
between the measured $\kappa_{CCN}$ and $\kappa_{HTDMA}$ values is likely attributed to the surfactant effect. It was
reported that organics matter in the particles could serve as surfactant and lower surface tension by
about 0.01-0.032 N m$^{-1}$, leading to a higher CCN activity and thus a higher $\kappa_{CCN}$ (Petters and
Kreidenweis, 2013;Ovadnevaite et al., 2017;Liu et al., 2018). According to Eqs. (4) and (5), the



$\kappa_{CCN}$ was more susceptibly affected by the value of surface tension than that of $\kappa_{HTDMA}$, which would
lead to the discrepancy between $\kappa_{CCN}$ and $\kappa_{HTDMA}$ values. Note that surface tension effect is not the
only factor which leads to a higher $\kappa_{CCN}$. It was found that $\kappa_{CCN}$ could be higher than $\kappa_{HTDMA}$, since
the existence of the slightly soluble compounds inhibits water uptake under subsaturation conditions
(Zhao et al., 2016;Pajunoja et al., 2015;Dusek et al., 2011;Petters et al., 2009).
**3.2  The average size-resolved hygroscopicity and volatility of OA**

The composition of organics could vary on a large scale with diameters due to different sources

and aging processes, which would further affect their properties. Figure 3 presents the average size-
resolved hygroscopicity and volatility of OA. The $\kappa_{OA}$ values (vertical red lines in Fig. 3) ranged
from 0.058 to 0.09, within the range (0.05-0.15 at 100 nm) previously reported in the PRD region
(Hong et al., 2018) and slightly higher than that (0.03-0.06 at 250 nm) at a mountain site in Germany
(Wu et al., 2013). In general, the $\kappa_{OA}$ values increased with particle sizes from 0.058 at 30 nm to
0.09 at 150 and 200 nm, similar to the feature observed in urban and forest environments (Kim et
al., 2020;Deng et al., 2019). The increases of the $\kappa_{OA}$ values with particle sizes could be explained
by the oxidation level of organic aerosols (Massoli et al., 2010;Lambe et al., 2011;Xu et al., 2021).
Specifically, the hygroscopicity of OA was often found to be positively correlated to its oxidation
level (Mei et al., 2013;Lambe et al., 2011), which was usually represented by $f_{44}$, O/C ratio, or $\overline{OS_c}$.
Thus, the higher $\kappa_{OA}$ values at larger particle diameters in this study might correspond to a higher
aging degree of these particles, and this was confirmed by the increasing trend of $f_{44}$ with particle
diameters, i.e., the increasing fraction of $CO_2^+$ in OA in large particles (Fig. S8). Previous field
studies also indicated that $f_{44}$ increased with particle diameters (Kim et al., 2020;Cai et al., 2018),





leading to a higher $\kappa_{OA}$ value.
Besides the hygroscopicity of OA, we observed the size dependence of volatility. As shown in
Fig. 3, the mass fraction of ELVOA increases from 0.16 to 0.30 with the particle diameter, indicating
that the particles could be more aged at larger diameters, consistent with the higher $\kappa_{OA}$ values as
discussed above. The ELVOA fraction in this campaign was higher than that in Beijing in summer
(0.13) measured by a thermodenuder (TD) coupled to an AMS (Xu et al., 2019), but similar to that
in Athens (0.3) using a similar TD system (Louvaris et al., 2017). The SVOA generally contributed
42%-57% to the OA at all measured sizes, comparable to the values reported in Centreville and
Raleigh (66-75%, Saha et al., 2017), Beijing (64%, Xu et al., 2019) and Mexico City (39%-73%,
Cappa and Jimenez, 2010). Note that the relationship between volatility and oxidation state of OA
is not usually strong. Saha et al. (2017) reported weak correlations (R < 0.3) between the mean
volatility ($\overline{C^*}$) and the mean oxidation state ($\overline{OS_C}$). Hong et al. (2017) also found that the volatility
distribution of OA derived from the combined V-TDMA and evaporation dynamic model could not
be fully explained by the organic factions determined by the PMF analysis based on the AMS data.
This is probably because the volatility was not only dependent on the $\overline{OS_C}$, but also the number of
atomic carbon (Donahue et al., 2011). In spite of this, the size-resolved volatility distribution can
provide a rough estimate of the aging degree of OA.
**3.3 The diurnal variation of OA hygroscopicity and volatility**
As discussed in Sect. 3.2, the hygroscopicity and volatility of OA could vary on a large range
with particle diameters, which might be attributed to photochemical reactions and the OA sources.
In this section, the diurnal variation of hygroscopicity and volatility of OA at different particle sizes



was investigated, in combination with the PMF results. In general, the mass fraction of organics
showed an obvious diurnal pattern during the whole campaign, with two peaks at about 14:00 and
19:00 LT (Fig. 4a), implying significant impacts of photochemical reactions and local emissions.
Based on the PMF results (Fig. 5), the afternoon peak was attributed to secondary organics aerosol
(SOA) formation (aBBOA and LOOA) during daytime, while the evening peak was explained by
local residential activity (e.g., biomass burning and cooking, HOA and BBOA), as will be discussed
later. A similar late-afternoon peak was observed in Hong Kong (Lee et al., 2013), where the OA
enhancement was mainly contributed by traffic emissions. The $f_{44}$ remained at a high level during
daytime, consistent with strong photochemical reactions. A similar diurnal pattern was observed in
the urban and sub-urban regions (Hong et al., 2018;Hu et al., 2016;Thalman et al., 2017), suggesting
the consistent aging processes of pre-existing OA. In contrast, Deng et al. (2019) reported a relative
low OA oxidation state during daytime in a forest environment, which could be explained by the
SOA formation through photochemical oxidation of BVOCs.

The calculated $\kappa_{HTDMA}$ and $\kappa_{AMS}$ (the blue and red lines in Fig. 4b, respectively) values at 200

nm based on Eqs. (4) and (7) both reached minimum during daytime which was consistent with high
OA fractions. This may be explained by lower hygroscopic of OA than inorganics as found in
previous studies (Pajunoja et al., 2015;Zhao et al., 2015; Kuang et al., 2020b) as well as the low $\kappa_i$
values shown in Table 1. Although OA in a higher oxidation state could be hydrophilic (Massoli et
al., 2010), the primary OA is usually considered to be hydrophobic substance and their mixture
would be less hygroscopic (usually with average $\kappa$ =0.1). The $\kappa_{AMS}$ values were generally consistent
with those of the $\kappa_{HTDMA}$ during daytime while the overestimated $\kappa_{OA}$ was observed during nighttime.
This implies a lower $\kappa_{OA}$ value than 0.1 at 200 nm during the nighttime, probably due to less



oxidation processes at night than those under the sunlight.
The average diurnal profile of PNSD is shown in Fig. 4c. Besides a stable accumulation mode
peaked at around 100 nm, a significantly growing mode of particle number from 20nm to 80 nm
was observed from 12:00 to 20:00 LT, which could be attributed to the frequently occurred NPF
during the campaign (Fig. 1a).
The size-resolved diurnal variations of $\kappa_{OA}$ was explored in Fig. 6. Note that the $\kappa_{OA}$ values are
presented in 2-hour resolution due to the low data coverage (Figs. 1 and 2). In general, a significantly
different pattern was observed between Aitken mode and accumulation mode. For Aitken mode
particles (30-100 nm), the $\kappa_{OA}$ values were higher (0.05-0.1) before dawn than those (0.02-0.07)
during daytime, while this trend began to overturn at 150 and 200 nm, where the $\kappa_{OA}$ values peaked
at noon (~0.09, Fig. 6). As reported in literature, the hygroscopicity of organics was partly dependent
on the aging degree (Liu et al., 2021;Zhao et al., 2016;Kim et al., 2020). The diurnal characteristics
of the size-resolved $\kappa_{OA}$ indicate that the OA in small particles (30-100 nm) was fresh and became
aged in large particles. For the same campaign, Kuang et al. (2021) showed a negative correlation
(R=-0.25) between LOOA and $\kappa_{OA}$, while a positive correlation (R=0.35) between aBBOA and $\kappa_{OA}$.
Thus, the decrease of $\kappa_{OA}$ for Aitken mode particles during daytime might be attributed to the
daytime formation of LOOA through gas-particle partitioning (Fig. 5). A similar phenomenon was
reported by Deng et al. (2019) in a forest environment, which might be attributed to the
photochemical reactions of BVOCs. Therefore, OA in small particles might be less aged and was
primarily contributed by photochemical oxidation of VOCs. In contrast, it is likely that the
accumulation mode particles became aged through photochemical oxidation during daylight, as
evidenced by higher fractions of ELVOA at 200 nm and higher $\kappa_{OA}$ (Figs. 6 and 7) during daytime.

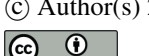



According to the PMF analysis, the daytime formation of aBBOA likely resulted from the aging
processes of primary OA or biomass burning related precursors (Fig. 5). As suggested by Kuang et
al. (2021), the daytime formation of aBBOA (Fig. 5) would lead to an increase of $\kappa_{OA}$, which likely
explained the noontime $\kappa_{OA}$ peak at 150 and 200 nm. It suggested that the OA in the accumulation
mode was more influenced by the aging processes through photochemical reactions (leading to
aBBOA formation).
The average size-resolved volatility distribution of OA during daytime (8:00 to 16:00 LT) and
nighttime (20:00 to 4:00 LT) was demonstrated in Fig. 7. A higher fraction of semi-volatile organic
aerosol (SVOA) was observed at six measured sizes (30, 50, 80, 100, 150, and 200 nm) during
daytime. SVOA was usually related to primary emission (e.g., traffic, biomass burning) and gas-
particle partitioning (Donahue et al., 2012;Jathar et al., 2020;Hong et al., 2017; Saha et al., 2017).
Two primary emission factors, BBOA and HOA, remained at a relative low level during daytime,
suggesting that the higher fraction of SVOA during daylight might be more originated from gas-
particle partitioning. Note that gas-particle partitioning (leading to LOOA formation) could occur
at all measured diameters, as shown by the higher daytime fractions of SVOA (Fig. 7). In summary,
the above results indicate that the negative effect of LOOA on $\kappa_{OA}$ might exist at all diameters, while
the positive effect of aBBOA was more dominant at larger particle sizes.
Meanwhile, the decreasing trend of $\kappa_{OA}$ was observed from 18:00 to 24:00 at 80 and 100 nm
which might be related to the high mass fraction of OA from primary emissions (HOA and BBOA,
Fig. 5), owing to their hydrophobic nature. These two primary factors were associated with traffic
emissions, cooking and biomass burning. Zhang et al. (2005b) constructed the size distribution of
HOA based on the size-resolved m/z 44 and 57 from the AMS measurement and showed that HOA





was dominant (~75%) in ultrafine particles ($D_{va}$<100 nm). The size-resolved PMF results from Sun
et al. (2012) also indicated a high mass fraction of HOA (0.3-0.4) in Aitken mode particles. The
mass distribution of BC could be used to represent the distribution of primary OA (Cubison et al.,
2008;Wang et al., 2010;Zhang et al., 2005a) due to similar source origins for BC and HOA/BBOA.
The average mass fraction of BC peaked at about 80-100 nm (Fig. S6a), suggesting that HOA and
BBOA might be dominant at this size range. The BC peaks at 80 nm and 100 nm were consistent
with those of the SVOA mass fraction (Fig. 3), which was attributed to biomass burning as similar
characteristics for the BC peak were shown in other studies (May et al., 2013;Huffman et al.,
2009;Donahue et al., 2011). Furthermore, this conclusion was supported by the hygroscopicity
measurements as a significant NH mode for 80-100 nm particles was found (Fig. 2). Overall, these
results highlight that the diurnal variation of physicochemical properties of OA could vary in a large
range with particle diameters, and further investigation is needed.
**3.4 Implication for CCN activity**

The CCN activity and its prediction is essential in global climate model and evaluation. A $\kappa_{OA}$

value of 0.1~0.15 was widely adopted in the prediction of $N_{CCN}$ based on aerosol chemical
composition (Meng et al., 2014;Wang et al., 2010;Almeida et al., 2014). As discussed in Sect. 3.3,
the $\kappa_{OA}$ values might be dependent on particle sizes and vary diurnally, which in turn affect $N_{CCN}$.
Here, different $\kappa_{OA}$ values were adopted to predict $N_{CCN}$ and the impact of $\kappa_{OA}$ on $N_{CCN}$ was
investigated through comparison between the predicted and measured $N_{CCN}$. Note that we only
discussed the $N_{CCN}$ at 0.1%, 0.2%, 0.4% and 0.7% SS, since the $D_{50}$ at higher SS (0.9% and 1.0%)
was within a narrow range (35-60 nm).



The $N_{CCN}$ at a certain SS can be calculated using PNSD and $D_{50}$:
$N_{CCN,p}(SS) = \int_{D_{50}}^{\infty} n_i dlogDp_i$                            (16)
where $n_i$ is the particle distribution function at $Dp_i$ and $D_{50}$ is determined from the $\kappa_{AMS}$ using Eqs.
(5) and (7). The $D_{50}$ at 0.1%, 0.2%, 0.4% and 0.7% SS ranged from about 130-160 nm, 90-110 nm,
60-80 nm and 45-60 nm, respectively. Three $\kappa_{OA}$ schemes were proposed to predict $N_{CCN}$: (1) fixed
$\kappa_{OA}$, where $\kappa_{OA}$ was assumed to be 0.1 for all size particles. (2) size-resolved $\kappa_{OA}$ (SR $\kappa_{OA}$), where $\kappa_{OA}$
$\kappa_{OA}$ was taken from average size-resolved $\kappa_{OA}$ ($\kappa_{OA}$ at 50, 80, 100 and 150 nm for 0.7%, 0.4%, 0.2%
and 0.1% SS, respectively) in Sect. 3.2. (3) size-resolved diurnal $\kappa_{OA}$ (SR diurnal $\kappa_{OA}$), where $\kappa_{OA}$
was the average diurnal value of $\kappa_{OA}$ at each diameter ($\kappa_{OA}$ at 50, 80, 100 and 150 nm for 0.7%,
0.4%, 0.2% and 0.1% SS, respectively) as shown in Sect. 3.3. The $\kappa_{AMS}$ was calculated based on the
chemical composition at the corresponding $D_{50}$ range. Note that the $N_{CCN}$ prediction based on the
SR diurnal $\kappa_{OA}$ scheme was presented in 2 h time resolution and the particles were assumed to be
internally mixed in Eq. (16). Cai et al. (2018) compared different approaches in predicting $N_{CCN}$ and
found that mixing state assumption played a minor role in the prediction, while the surfactant effect
should be taken into account. As aforementioned, organics can increase the CCN activity by
decreasing surface tension, which might lead to significant discrepancy between $\kappa_{HTDMA}$ and $\kappa_{CCN}$
in this campaign (Fig. S7). In addition, this effect could result in a significant underestimation of
$N_{CCN}$ (Ovadnevaite et al., 2017;Liu et al., 2018;Good et al., 2010; Noziere, 2016).
Here, we evaluate the surface tension effect by comparing $\kappa_{HTDMA}$ and $\kappa_{CCN}$ as a function of
$\sigma_{s/a}$ (Fig. S9). The $\kappa_{CCN}$ reached $\kappa_{HTDMA}$ when the $\sigma_{s/a}$ values were set to be about 0.059 N m$^{-1}$ at
0.7%, 0.9% and 1.0% SS, 0.053 N m$^{-1}$ at 0.4% SS, 0.047 N m$^{-1}$ at 0.2% SS, and 0.049 N m$^{-1}$ at 0.1%
SS, respectively. Thus, we adopted $\sigma_{s/a}$ values of 0.049, 0.047, 0.053 and 0.059 N m$^{-1}$ to predict





$N_{CCN}$ at 0.1%, 0.2%, 0.4% and 0.7% SS, respectively. In general, the $N_{CCN}$ prediction could be
significantly improved by considering the surfactant effect (Fig. S10).

The deviation of the $N_{CCN}$ prediction ($\delta_{N_{CCN}}$) at a certain SS is defined as (Cai et al., 2021b):

$\delta_{N_{CCN}}(SS) = \frac{N_{CCN,m}(SS) - N_{CCN,p}(SS)}{N_{CCN,m}(SS)} 100\%$                    (17)
where $N_{CCN,m}(SS)$ is the measured $N_{CCN}$ at a specific SS. A negative $\delta_{N_{CCN}}$ indicates an
overestimate of $N_{CCN}$, and vice versa.

Figure 8 shows the $\delta_{N_{CCN}}$ at different SS for the three $\kappa_{OA}$ schemes. Fixed $\kappa_{OA}$ scheme gave

generally a negative value of $\delta_{N_{CCN}}$ (-0.18 to -0.02) at 0.7% SS, indicating an $N_{CCN}$ overestimation,
due to lower $\kappa_{OA}$ values for smaller particles. A significant diurnal pattern of $\delta_{N_{CCN}}$ was observed at
all SS. The $\delta_{N_{CCN}}$ was relatively higher during daytime at 0.1% SS, while an opposite pattern was
shown at high SS, consistent with the size-dependent variation of $\kappa_{OA}$ (Fig. 6). Hence, the fixed $\kappa_{OA}$
scheme could lead to an obvious discrepancy in the $N_{CCN}$ prediction as SS increased. The results
based on the SR $\kappa_{OA}$ scheme showed that the minimum $\delta_{N_{CCN}}$ value at 0.7% SS increased from -
0.18 in the fixed $\kappa_{OA}$ scheme to -0.08, indicating the improvement for the $N_{CCN}$ prediction at high
SS (Fig. 8b). However, only minor improvement was observed at SS lower than 0.4 % because of
the low employed $\kappa_{OA}$ (about 0.08), which was close to the $\kappa_{OA}$ value (0.1) adopted in the fixed $\kappa_{OA}$
scheme. A significant difference of $\delta_{N_{CCN}}$ was still observed in the diurnal pattern at high and low
SS, implying the impact of the diurnal variation of $\kappa_{OA}$ on the $N_{CCN}$ prediction. To further investigate
this impact, the SR diurnal $\kappa_{OA}$ scheme was employed to calculate $\delta_{N_{CCN}}$ and the results were shown
in Fig. 8c. The $\delta_{N_{CCN}}$ value at 0.7% SS varied from -0.04 to 0.09 with an average value of 0, whereas
it ranged from 0 to 0.11 at 0.1% SS. Hence, the discrepancies of $\delta_{N_{CCN}}$ among different SS became
minor compared to the other two schemes as a relatively flat diurnal pattern of $\delta_{N_{CCN}}$ was observed



at all SS. It implies that better prediction of $N_{CCN}$ could be achieved by considering the diurnal
variation and the size dependence of $\kappa_{OA}$.
**4.    Conclusions**
A rural field measurement was conducted at the Heshan supersite in the PRD region of China
during October and November 2019. We investigated the diurnal variation and size dependence in
the hygroscopicity and volatility of OA in combination with the PMF analysis of the AMS data. The
impacts of OA on the CCN number concentration at different SS were discussed for various given
size-dependent $\kappa_{OA}$ values.
In general, the average $\kappa_{OA}$ values varied from 0.058 at 30 nm to 0.09 at 200 nm, indicating a
higher oxidation degree of OA at larger sizes than at smaller sizes. This is consistent with particle
volatility: the mass fraction of ELVOA increased (0.16-0.30) with increasing particle diameters.
Our results suggest that the formation and aging processes of OA might vary with particle sizes.
An oppositely diurnal pattern of $\kappa_{OA}$ was observed between Aitken mode (30-100 nm) and
Accumulation mode (150 and 200 nm) particles, suggesting different atmospheric evolution
processes of OA at different diameters. The gas-particle partitioning could decrease the $\kappa_{OA}$, while
the aging processes of preexisting particles could enhance the hygroscopicity of OA. The $\kappa_{OA}$ values
for 30-100 nm particles reached minimal (0.02-0.07) and a high $\kappa_{OA}$ value (~0.09) for 150 and 200
nm particles was observed during daytime, suggesting that the aging processes of preexisting
particles were more dominant at accumulation mode particles. In addition, the mass faction of
SVOA was higher during daytime at all measured diameters, implying that the formation of LOOA



through gas-particle partitioning was independent of particle diameters.
The impact of the size-resolved diurnal variation of $\kappa_{OA}$ on the $N_{CCN}$ was investigated. The use
of fixed $\kappa_{OA}$ ($\kappa_{OA}$=0.1) overestimated the $N_{CCN}$ up to 18% at 0.7% SS. The diurnal deviation became
obvious at 0.7% SS and minor at 0.1% SS during daytime, owing to the size-dependent variation of
$\kappa_{OA}$. The $N_{CCN}$ prediction at 0.7% SS was improved if the SR $\kappa_{OA}$ scheme was used, while the
diurnal variation of $\delta_{N_{CCN}}$ still existed. Better predictions can be obtained by using SR diurnal $\kappa_{OA}$.
Our results highlight that the physical properties of OA can vary in a large range at different size
ranges due to the formation and aging processes, and the size-resolved diurnal variation in $\kappa_{OA}$ plays
an important role in the $N_{CCN}$ prediction at different SS. Further studies on the size-resolved
physicochemical properties of OA should be performed in different environments to better
understand their impact on cloud formation and hence climate.

*Data availability.* Data from the measurements are available at
https://doi.org/10.6084/m9.figshare.18094277.v1 (Cai et al., 2022).

*Supplement.* The supplement related to this article is available online at xxx.

*Author contributions.* **MC, SH, BY and LL** designed the research. **MC, SH, MS, BY, YP, ZW, DC,**
**WC, QS, WL, BL and QS** performed the measurements. **MC, SH, BL, QS, LL, BY, WH, WC,**
**QS, WL, YP, ZW, HT, HX, FL, DX, TD, JS and JZ** analyzed the data. **MC, SH** and **LL** wrote the
paper with contributions from all co-authors.



*Competing interests.* The authors declare that they have no conflict of interest.

*Financial support.* This work was supported by the Key-Area Research and Development Program
of Guangdong Province (grant no. 2019B110206001), the National Key R&D Plan of China (grant
no. 2019YFE0106300 and 2018YFC0213904), the National Natural Science Foundation of China
(grant nos. 41877302, 91644225, 41775117 and 41807302), Guangdong Natural Science Funds for
Distinguished Young Scholar (grant no. 2018B030306037), Guangdong Innovative and
Entrepreneurial Research Team Program (grant no. 2016ZT06N263), Guangdong Province Key
Laboratory for Climate Change and Natural Disaster Studies (grant no. 2020B1212060025),
Guangdong Basic and Applied Basic Research Foundation (grant nos. 2019A1515110790 and
2019A1515110791), Science and Technology Research project of Guangdong Meteorological
Bureau (grant no. GRMC2018M07), the Natural Science Foundation of Guangdong Province,
China (grant no. 2016A030311007), Science and Technology Innovation Team Plan of Guangdong
Meteorological Bureau (grant no. GRMCTD202003), and Science and Technology Program of
Guangdong    Province    (Science    and    Technology    Innovation    Platform    Category,    No.
2019B121201002).

*Acknowledgements.* Additional support from the crew of the Heshan supersite and Guangdong
Environmental Monitoring Center is greatly acknowledged.



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

Contributions to CCN Concentrations Over a Midlatitude Forest in Japan, J. Geophys. Res. Atmos.,

123, 9703-9723, 10.1029/2017JD027292, 2018.

Deng, Y., Yai, H., Fujinari, H., Kawana, K., Nakayama, T., and Mochida, M.: Diurnal variation

and size dependence of the hygroscopicity of organic aerosol at a forest site in Wakayama, Japan:

their relationship to CCN concentrations, Atmos. Chem. Phys., 19, 5889-5903, 10.5194/acp-19-

5889-2019, 2019.

Donahue, N. M., Epstein, S. A., Pandis, S. N., and Robinson, A. L.: A two-dimensional

volatility basis set: 1. organic-aerosol mixing thermodynamics, Atmos. Chem. Phys., 11, 3303-3318,

10.5194/acp-11-3303-2011, 2011.

Donahue, N. M., Kroll, J. H., Pandis, S. N., and Robinson, A. L.: A two-dimensional volatility

basis set – Part 2: Diagnostics of organic-aerosol evolution, Atmos. Chem. Phys., 12, 615-634,

10.5194/acp-12-615-2012, 2012.

Dusek, U., Frank, G. P., Massling, A., Zeromskiene, K., Iinuma, Y., Schmid, O., Helas, G.,

Hennig, T., Wiedensohler, A., and Andreae, M. O.: Water uptake by biomass burning aerosol at sub-

and supersaturated conditions: closure studies and implications for the role of organics, Atmos.

Chem. Phys., 11, 9519-9532, 10.5194/acp-11-9519-2011, 2011.

Engelhart, G. J., Moore, R. H., Nenes, A., and Pandis, S. N.: Cloud condensation nuclei activity

of isoprene secondary organic aerosol, J. Geophys. Res. Atmos., 116,

https://doi.org/10.1029/2010JD014706, 2011.

Epstein, S. A., Riipinen, I., and Donahue, N. M.: A Semiempirical Correlation between

Enthalpy of Vaporization and Saturation Concentration for Organic Aerosol, Environ. Sci. Technol.,

44, 743-748, 10.1021/es902497z, 2010.

Gantt, B., and Meskhidze, N.: The physical and chemical characteristics of marine primary

organic aerosol: a review, Atmos. Chem. Phys., 13, 3979-3996, 10.5194/acp-13-3979-2013, 2013.

Good, N., Topping, D., Allan, J., Flynn, M., Fuentes, E., Irwin, M., Williams, P., Coe, H., and

McFiggans, G.: Consistency between parameterisations of aerosol hygroscopicity and CCN activity

during the RHaMBLe discovery cruise, Atmos. Chem. Phys., 10, 3189-3203, 2010.





Guo, J., Zhou, S., Cai, M., Zhao, J., Song, W., Zhao, W., Hu, W., Sun, Y., He, Y., Yang, C., Xu,
X., Zhang, Z., Cheng, P., Fan, Q., Hang, J., Fan, S., Wang, X., and Wang, X.: Characterization of
submicron particles by time-of-flight aerosol chemical speciation monitor (ToF-ACSM) during
wintertime: aerosol composition, sources, and chemical processes in Guangzhou, China, Atmos.
Chem. Phys., 20, 7595-7615, 10.5194/acp-20-7595-2020, 2020.
Gysel, M., Crosier, J., Topping, D. O., Whitehead, J. D., Bower, K. N., Cubison, M. J., Williams,
P. I., Flynn, M. J., McFiggans, G. B., and Coe, H.: Closure study between chemical composition
and hygroscopic growth of aerosol particles during TORCH2, Atmos. Chem. Phys., 7, 6131-6144,
10.5194/acp-7-6131-2007, 2007.
Hallquist, M., Wenger, J. C., Baltensperger, U., Rudich, Y., Simpson, D., Claeys, M., Dommen,
J., Donahue, N., George, C., and Goldstein, A.: The formation, properties and impact of secondary
organic aerosol: current and emerging issues, Atmospheric chemistry and physics, 9, 5155-5236,

2009.

Hong, J., Äijälä, M., Häme, S. A. K., Hao, L., Duplissy, J., Heikkinen, L. M., Nie, W., Mikkilä,
J., Kulmala, M., Prisle, N. L., Virtanen, A., Ehn, M., Paasonen, P., Worsnop, D. R., Riipinen, I.,
Petäjä, T., and Kerminen, V. M.: Estimates of the organic aerosol volatility in a boreal forest using
two independent methods, Atmos. Chem. Phys., 17, 4387-4399, 10.5194/acp-17-4387-2017, 2017.
Hong, J., Xu, H., Tan, H., Yin, C., Hao, L., Li, F., Cai, M., Deng, X., Wang, N., Su, H., Cheng,
Y., Wang, L., Petäjä, T., and Kerminen, V. M.: Mixing state and particle hygroscopicity of organic-
dominated aerosols over the Pearl River Delta region in China, Atmos. Chem. Phys., 18, 14079-
14094, 10.5194/acp-18-14079-2018, 2018.
Hu, W., Hu, M., Hu, W., Jimenez, J. L., Yuan, B., Chen, W., Wang, M., Wu, Y., Chen, C., Wang,
Z., Peng, J., Zeng, L., and Shao, M.: Chemical composition, sources, and aging process of
submicron aerosols in Beijing: Contrast between summer and winter, J. Geophys. Res. Atmos., 121,
1955-1977, https://doi.org/10.1002/2015JD024020, 2016.
Huang, R.-J., Zhang, Y., Bozzetti, C., Ho, K.-F., Cao, J.-J., Han, Y., Daellenbach, K. R., Slowik,
J. G., Platt, S. M., and Canonaco, F.: High secondary aerosol contribution to particulate pollution
during haze events in China, Nature, 514, 218, 2014.
Huang, S., Wu, Z., Poulain, L., van Pinxteren, M., Merkel, M., Assmann, D., Herrmann, H.,



and Wiedensohler, A.: Source apportionment of the organic aerosol over the Atlantic Ocean from
53°N to 53°S: significant contributions from marine emissions and long-range transport, Atmos.
Chem. Phys., 18, 18043-18062, 10.5194/acp-18-18043-2018, 2018.
Huffman, J. A., Docherty, K. S., Aiken, A. C., Cubison, M. J., Ulbrich, I. M., DeCarlo, P. F.,
Sueper, D., Jayne, J. T., Worsnop, D. R., Ziemann, P. J., and Jimenez, J. L.: Chemically-resolved
aerosol volatility measurements from two megacity field studies, Atmos. Chem. Phys., 9, 7161-
7182, 10.5194/acp-9-7161-2009, 2009.
Jathar, S. H., Sharma, N., Galang, A., Vanderheyden, C., Takhar, M., Chan, A. W. H., Pierce, J.
R., and Volckens, J.: Measuring and modeling the primary organic aerosol volatility from a modern
non-road diesel engine, Atmos. Environ., 223, 117221,
https://doi.org/10.1016/j.atmosenv.2019.117221, 2020.
Jiang, R., Tan, H., Tang, L., Cai, M., Yin, Y., Li, F., Liu, L., Xu, H., Chan, P. W., and Deng, X.:
Comparison of aerosol hygroscopicity and mixing state between winter and summer seasons in Pearl
River Delta region, China, Atmos. Res., 169, 160-170, 2016.
Jimenez, J. L., Canagaratna, M., Donahue, N., Prevot, A., Zhang, Q., Kroll, J. H., DeCarlo, P.
F., Allan, J. D., Coe, H., and Ng, N.: Evolution of organic aerosols in the atmosphere, Science, 326,

1525-1529, 2009.

Kanakidou, M., Seinfeld, J., Pandis, S., Barnes, I., Dentener, F., Facchini, M., Dingenen, R. V.,
Ervens, B., Nenes, A., and Nielsen, C.: Organic aerosol and global climate modelling: a review,
Atmos. Chem. Phys., 5, 1053-1123, 2005.
Kim, N., Yum, S. S., Park, M., Park, J. S., Shin, H. J., and Ahn, J. Y.: Hygroscopicity of urban
aerosols and its link to size-resolved chemical composition during spring and summer in Seoul,
Korea, Atmos. Chem. Phys., 20, 11245-11262, 10.5194/acp-20-11245-2020, 2020.
Kiyoura, R., and Urano, K.: Mechanism, Kinetics, and Equilibrium of Thermal Decomposition
of Ammonium Sulfate, Ind. Eng. Chem. Process Des. Dev. , 9, 489-494, 10.1021/i260036a001,

1970.

Kuang, Y., He, Y., Xu, W., Yuan, B., Zhang, G., Ma, Z., Wu, C., Wang, C., Wang, S., Zhang,
S., Tao, J., Ma, N., Su, H., Cheng, Y., Shao, M., and Sun, Y.: Photochemical Aqueous-Phase
Reactions Induce Rapid Daytime Formation of Oxygenated Organic Aerosol on the North China



Plain, Environ. Sci. Technol., 54, 3849-3860, 10.1021/acs.est.9b06836, 2020a.
Kuang, Y., Xu, W., Tao, J., Ma, N., Zhao, C., and Shao, M.: A Review on Laboratory Studies
and Field Measurements of Atmospheric Organic Aerosol Hygroscopicity and Its Parameterization
Based on Oxidation Levels, Curr. Pollut. Rep., 10.1007/s40726-020-00164-2, 2020b.
Kuang, Y., Huang, S., Xue, B., Luo, B., Song, Q., Chen, W., Hu, W., Li, W., Zhao, P., Cai, M.,
Peng, Y., Qi, J., Li, T., Wang, S., Chen, D., Yue, D., Yuan, B., and Shao, M.: Contrasting effects of
secondary organic aerosol formations on organic aerosol hygroscopicity, Atmos. Chem. Phys., 21,
10375-10391, 10.5194/acp-21-10375-2021, 2021.
Lambe, A. T., Onasch, T. B., Massoli, P., Croasdale, D. R., Wright, J. P., Ahern, A. T., Williams,
L. R., Worsnop, D. R., Brune, W. H., and Davidovits, P.: Laboratory studies of the chemical
composition and cloud condensation nuclei (CCN) activity of secondary organic aerosol (SOA) and
oxidized primary organic aerosol (OPOA), Atmos. Chem. Phys., 11, 8913-8928, 10.5194/acp-11-

8913-2011, 2011.

Lee, B.-H., Kostenidou, E., Hildebrandt, L., Riipinen, I., Engelhart, G., Mohr, C., DeCarlo, P.,
Mihalopoulos, N., Prevot, A., Baltensperger, U.: Measurement of the ambient organic aerosol
volatility distribution: application during the Finokalia Aerosol Measurement Experiment (FAME-
2008), Atmos. Chem. Phys., 10, 12149-12160, 2010.
Lee, B.-H., Pierce, J. R., Engelhart, G. J., and Pandis, S. N.: Volatility of secondary organic
aerosol from the ozonolysis of monoterpenes, Atmos. Environ., 45, 2443-2452, 2011.
Lee, B. P., Li, Y. J., Yu, J. Z., Louie, P. K., and Chan, C. K.: Physical and chemical
characterization of ambient aerosol by HR-ToF-AMS at a suburban site in Hong Kong during
springtime 2011, J. Geophys. Res. Atmos., 118, 8625-8639, 2013.
Li, J., Wang, G., Zhou, B., Cheng, C., Cao, J., Shen, Z., and An, Z.: Airborne particulate
organics at the summit (2060m, a.s.l.) of Mt. Hua in central China during winter: Implications for
biofuel    and    coal    combustion,    Atmos.    Res.,    106,    108-119,
https://doi.org/10.1016/j.atmosres.2011.11.012, 2012.
Li, Y. J., Lee, B., Yu, J., Ng, N., and Chan, C. K.: Evaluating the degree of oxygenation of
organic aerosol during foggy and hazy days in Hong Kong using high-resolution time-of-flight
aerosol mass spectrometry (HR-ToF-AMS), Atmos. Chem. Phys., 13, 8739-8753, 2013.





Liu, J., Zhang, F., Xu, W., Sun, Y., Chen, L., Li, S., Ren, J., Hu, B., Wu, H., and Zhang, R.:
Hygroscopicity of Organic Aerosols Linked to Formation Mechanisms, Geophys. Res. Lett., 48,
e2020GL091683, https://doi.org/10.1029/2020GL091683, 2021.
Liu, P., Song, M., Zhao, T., Gunthe, S. S., Ham, S., He, Y., Qin, Y. M., Gong, Z., Amorim, J.
C., Bertram, A. K., and Martin, S. T.: Resolving the mechanisms of hygroscopic growth and cloud
condensation nuclei activity for organic particulate matter, Nat. Commun., 9, 4076,
10.1038/s41467-018-06622-2, 2018.
Liu, P. F., Zhao, C. S., Göbel, T., Hallbauer, E., Nowak, A., Ran, L., Xu, W. Y., Deng, Z. Z.,
Ma, N., Mildenberger, K., Henning, S., Stratmann, F., and Wiedensohler, A.: Hygroscopic properties
of aerosol particles at high relative humidity and their diurnal variations in the North China Plain,
Atmos. Chem. Phys., 11, 3479-3494, 10.5194/acp-11-3479-2011, 2011.
Liu, X., and Wang, J.: How important is organic aerosol hygroscopicity to aerosol indirect
forcing?, Environ. Res. Lett., 5, 044010, 10.1088/1748-9326/5/4/044010, 2010.
Louvaris, E. E., Florou, K., Karnezi, E., Papanastasiou, D. K., Gkatzelis, G. I., and Pandis, S.
N.: Volatility of source apportioned wintertime organic aerosol in the city of Athens, Atmos.
Environ., 158, 138-147, https://doi.org/10.1016/j.atmosenv.2017.03.042, 2017.
Noziere, B.: Don't forget the surface, Science, 351, 1396-1397, 10.1126/science.aaf3253, 2016.
Ma, N., Zhao, C., Tao, J., Wu, Z., Kecorius, S., Wang, Z., Größ, J., Liu, H., Bian, Y., and Kuang,
Y.: Variation of CCN activity during new particle formation events in the North China Plain, Atmos.
Chem. Phys., 16, 8593-8607, 2016.
Massling, A., Stock, M., and Wiedensohler, A.: Diurnal, weekly, and seasonal variation of
hygroscopic properties of submicrometer urban aerosol particles, Atmos. Environ., 39, 3911-3922,
10.1016/j.atmosenv.2005.03.020, 2005.
Massoli, P., Lambe, A., Ahern, A., Williams, L., Ehn, M., Mikkilä, J., Canagaratna, M., Brune,
W., Onasch, T., and Jayne, J.: Relationship between aerosol oxidation level and hygroscopic
properties of laboratory generated secondary organic aerosol (SOA) particles, Geophys. Res. Lett.,

37, 2010.

May, A. A., Levin, E. J. T., Hennigan, C. J., Riipinen, I., Lee, T., Collett Jr., J. L., Jimenez, J.
L., Kreidenweis, S. M., and Robinson, A. L.: Gas-particle partitioning of primary organic aerosol



emissions: 3. Biomass burning, J. Geophys. Res. Atmos., 118, 11,327-311,338,
https://doi.org/10.1002/jgrd.50828, 2013.
Mei, F., Setyan, A., Zhang, Q., and Wang, J.: CCN activity of organic aerosols observed
downwind of urban emissions during CARES, Atmos. Chem. Phys., 13, 12155-12169, 2013.
Meng, J. W., Yeung, M. C., Li, Y. J., Lee, B. Y. L., and Chan, C. K.: Size-resolved cloud
condensation nuclei (CCN) activity and closure analysis at the HKUST Supersite in Hong Kong,
Atmos. Chem. Phys., 14, 10267-10282, 10.5194/acp-14-10267-2014, 2014.
Mochida, M., Kuwata, M., Miyakawa, T., Takegawa, N., Kawamura, K., and Kondo, Y.:
Relationship between hygroscopicity and cloud condensation nuclei activity for urban aerosols in
Tokyo, J. Geophys. Res., 111, D23204, 10.1029/2005jd006980, 2006.
Onasch, T. B., Trimborn, A., Fortner, E. C., Jayne, J. T., Kok, G. L., Williams, L. R., Davidovits,
P., and Worsnop, D. R.: Soot Particle Aerosol Mass Spectrometer: Development, Validation, and
Initial Application, Aerosol Sci. Tech., 46, 804-817, 10.1080/02786826.2012.663948, 2012.
Ovadnevaite, J., Zuend, A., Laaksonen, A., Sanchez, K. J., Roberts, G., Ceburnis, D., Decesari,
S., Rinaldi, M., Hodas, N., Facchini, M. C., Seinfeld, J. H., and O' Dowd, C.: Surface tension
prevails over solute effect in organic-influenced cloud droplet activation, Nature, 546, 637-641,
10.1038/nature22806, 2017.
Paatero, P., and Tapper, U.: Positive matrix factorization: A non-negative factor model with
optimal utilization of error estimates of data values, Environmetrics, 5, 111-126,
10.1002/env.3170050203, 1994.
Paatero, P.: Least squares formulation of robust non-negative factor analysis, Chemometr Intell
Lab, 37, 23-35, 10.1016/S0169-7439(96)00044-5, 1997.
Pajunoja, A., Lambe, A. T., Hakala, J., Rastak, N., Cummings, M. J., Brogan, J. F., Hao, L.,
Paramonov, M., Hong, J., and Prisle, N. L.: Adsorptive uptake of water by semisolid secondary
organic aerosols, Geophys. Res. Lett., 42, 3063-3068, 2015.
Park, S. H., Rogak, S. N., and Grieshop, A. P.: A Two-Dimensional Laminar Flow Model for
Thermodenuders Applied to Vapor Pressure Measurements, Aerosol Sci. Technol., 47, 283-293,

10.1080/02786826.2012.750711, 2013.

Petters, M., and Kreidenweis, S.: A single parameter representation of hygroscopic growth and





cloud condensation nucleus activity, Atmos. Chem. Phys., 7, 1961-1971, 2007.
Petters, M. D., Wex, H., Carrico, C. M., Hallbauer, E., Massling, A., McMeeking, G. R.,
Poulain, L., Wu, Z., Kreidenweis, S. M., and Stratmann, F.: Towards closing the gap between
hygroscopic growth and activation for secondary organic aerosol – Part 2: Theoretical approaches,
Atmos. Chem. Phys., 9, 3999-4009, 10.5194/acp-9-3999-2009, 2009.
Petters, M., and Kreidenweis, S.: A single parameter representation of hygroscopic growth and
cloud condensation nucleus activity–Part 3: Including surfactant partitioning, Atmos. Chem. Phys.,

13, 1081-1091, 2013.

Philippin, S., Wiedensohler, A., and Stratmann, F.: Measurements of non-volatile fractions of
pollution aerosols with an eight-tube volatility tandem differential mobility analyzer (VTDMA-8),
J. Aerosol Sci., 35, 185-203, http://dx.doi.org/10.1016/j.jaerosci.2003.07.004, 2004.
Qin, Y. M., Tan, H. B., Li, Y. J., Schurman, M. I., Li, F., Canonaco, F., Prévôt, A. S. H., and
Chan, C. K.: The role of traffic emissions in particulate organics and nitrate at a downwind site in
the periphery of Guangzhou, China, Atmos. Chem. Phys., 1-31, 2017.
Rastak, N., Pajunoja, A., Acosta Navarro, J. C., Ma, J., Song, M., Partridge, D. G., Kirkevåg,
A., Leong, Y., Hu, W. W., Taylor, N. F., Lambe, A., Cerully, K., Bougiatioti, A., Liu, P., Krejci, R.,
Petäjä, T., Percival, C., Davidovits, P., Worsnop, D. R., Ekman, A. M. L., Nenes, A., Martin, S.,
Jimenez, J. L., Collins, D. R., Topping, D. O., Bertram, A. K., Zuend, A., Virtanen, A., and Riipinen,
I.: Microphysical explanation of the RH-dependent water affinity of biogenic organic aerosol and
its importance for climate, Geophys. Res. Lett., 44, 5167-5177, 10.1002/2017GL073056, 2017.
Riipinen, I., Pierce, J. R., Donahue, N. M., and Pandis, S. N.: Equilibration time scales of
organic aerosol inside thermodenuders: Evaporation kinetics versus thermodynamics, Atmos.
Environ., 44, 597-607, https://doi.org/10.1016/j.atmosenv.2009.11.022, 2010.
Rose, D., Nowak, A., Achtert, P., Wiedensohler, A., Hu, M., Shao, M., Zhang, Y., Andreae, M.
O., and Pöschl, U.: Cloud condensation nuclei in polluted air and biomass burning smoke near the
mega-city Guangzhou, China – Part 1: Size-resolved measurements and implications for the
modeling of aerosol particle hygroscopicity and CCN activity, Atmos. Chem. Phys., 10, 3365-3383,
10.5194/acp-10-3365-2010, 2010.
Saha, P. K., Khlystov, A., Grieshop, A. P.: Determining aerosol volatility parameters using a



"Dual Thermodenuder" system: application to laboratory-generated organic aerosols, Aerosol Sci.
Tech., 49, 620-632, 2015.
Saha, P. K., Khlystov, A., Yahya, K., Zhang, Y., Xu, L., Ng, N. L., Grieshop, A. P.: Quantifying
the volatility of organic aerosol in the southeastern US, Atmos. Chem. Phys., 17, 501-520, 2017.
Saleh, R., Walker, J., and Khlystov, A.: Determination of saturation pressure and enthalpy of
vaporization of semi-volatile aerosols: The integrated volume method, J. Aerosol Sci.e, 39, 876-887,
https://doi.org/10.1016/j.jaerosci.2008.06.004, 2008.
Seinfeld, J. H., and Pandis, S. N.: Atmospheric chemistry and physics: from air pollution to
climate change, John Wiley & Sons, 2016.
Shrivastava, M., Cappa, C. D., Fan, J., Goldstein, A. H., Guenther, A. B., Jimenez, J. L., Kuang,
C., Laskin, A., Martin, S. T., Ng, N. L., Petaja, T., Pierce, J. R., Rasch, P. J., Roldin, P., Seinfeld, J.
H., Shilling, J., Smith, J. N., Thornton, J. A., Volkamer, R., Wang, J., Worsnop, D. R., Zaveri, R. A.,
Zelenyuk, A., and Zhang, Q.: Recent advances in understanding secondary organic aerosol:
Implications    for    global    climate    forcing,    Rev.    Geophys.,    55,    509-559,
https://doi.org/10.1002/2016RG000540, 2017.
Stokes, R., and Robinson, R.: Interactions in aqueous nonelectrolyte solutions. I. Solute-
solvent equilibria, J. Phys. Chem., 70, 2126-2131, 1966.
Stolzenburg, M. R., and McMurry, P. H.: Equations Governing Single and Tandem DMA
Configurations and a New Lognormal Approximation to the Transfer Function, Aerosol Sci. Tech.,

928    42, 421-432, 10.1080/02786820802157823, 2008.

Sun, Y. L., Zhang, Q., Schwab, J. J., Yang, T., Ng, N. L., and Demerjian, K. L.: Factor analysis
of combined organic and inorganic aerosol mass spectra from high resolution aerosol mass
spectrometer measurements, Atmos. Chem. Phys., 12, 8537-8551, 10.5194/acp-12-8537-2012,

2012.

Tan, H., Xu, H., Wan, Q., Li, F., Deng, X., Chan, P. W., Xia, D., and Yin, Y.: Design and
Application of an Unattended Multifunctional H-TDMA System, J. Atmos. Oceanic Tech., 30, 1136-
1148, 10.1175/JTECH-D-12-00129.1, 2013a.
Tan, H., Yin, Y., Gu, X., Li, F., Chan, P. W., Xu, H., Deng, X., and Wan, Q.: An observational
study of the hygroscopic properties of aerosols over the Pearl River Delta region, Atmos. Environ.,



77, 817-826, http://dx.doi.org/10.1016/j.atmosenv.2013.05.049, 2013b.
Thalman, R., Sá, S. S. d., Palm, B. B., Barbosa, H. M., Pöhlker, M. L., Alexander, M. L., Brito,
J., Carbone, S., Castillo, P., Day, D. A.: CCN activity and organic hygroscopicity of aerosols
downwind of an urban region in central Amazonia: seasonal and diel variations and impact of
anthropogenic emissions, Atmos. Chem. Phys., 17, 11779-11801, 2017.
Ulbrich, I. M., Canagaratna, M. R., Zhang, Q., Worsnop, D. R., and Jimenez, J. L.:
Interpretation of organic components from Positive Matrix Factorization of aerosol mass
spectrometric data, Atmos. Chem. Phys., 9, 2891-2918, 10.5194/acp-9-2891-2009, 2009.
Volkamer, R., Jimenez, J. L., Martini, F. S., Dzepina, K., Qi, Z., Salcedo, D., Molina, L. T.,
Worsnop, D. R., and Molina, M. J.: Secondary organic aerosol formation from anthropogenic air
pollution: Rapid and higher than expected, Geophys. Res. Lett., 33, 254-269, 2006.
Wang, D. S., and Hildebrandt Ruiz, L.: Chlorine-initiated oxidation of n-alkanes under high-
NOx conditions: insights into secondary organic aerosol composition and volatility using a
FIGAERO–CIMS, Atmos. Chem. Phys., 18, 15535-15553, 10.5194/acp-18-15535-2018, 2018.
Wang, J., Lee, Y. N., Daum, P. H., Jayne, J., and Alexander, M. L.: Effects of aerosol organics
on cloud condensation nucleus (CCN) concentration and first indirect aerosol effect, Atmos. Chem.
Phys., 8, 6325-6339, 10.5194/acp-8-6325-2008, 2008.
Wang, J., Cubison, M., Aiken, A., Jimenez, J., and Collins, D.: The importance of aerosol
mixing state and size-resolved composition on CCN concentration and the variation of the
importance with atmospheric aging of aerosols, Atmos. Chem. Phys., 10, 7267-7283, 2010.
Wu, Z. J., Poulain, L., Henning, S., Dieckmann, K., Birmili, W., Merkel, M., van Pinxteren,
D., Spindler, G., Müller, K., Stratmann, F., Herrmann, H., and Wiedensohler, A.: Relating particle
hygroscopicity and CCN activity to chemical composition during the HCCT-2010 field campaign,
Atmos. Chem. Phys., 13, 7983-7996, 10.5194/acp-13-7983-2013, 2013.
Xu, W., Chen, C., Qiu, Y., Xie, C., Chen, Y., Ma, N., Xu, W., Fu, P., Wang, Z., Pan, X., Zhu, J.,
Ng, N. L., and Sun, Y.: Size-resolved characterization of organic aerosol in the North China Plain:
new insights from high resolution spectral analysis, Environ. Sci. Atmos., 1, 346-358,
10.1039/D1EA00025J, 2021.
Xu, W., Xie, C., Karnezi, E., Zhang, Q., Wang, J., Pandis, S. N., Ge, X., Zhang, J., An, J., Wang,





Q., Zhao, J., Du, W., Qiu, Y., Zhou, W., He, Y., Li, Y., Li, J., Fu, P., Wang, Z., Worsnop, D. R., and
Sun, Y.: Summertime aerosol volatility measurements in Beijing, China, Atmos. Chem. Phys., 19,
10205-10216, 10.5194/acp-19-10205-2019, 2019.
Yuan, L., Zhang, X., Feng, M., Liu, X., Che, Y., Xu, H., Schaefer, K., Wang, S., and Zhou, Y.:
Size-resolved hygroscopic behaviour and mixing state of submicron aerosols in a megacity of the
Sichuan Basin during pollution and fireworks episodes, Atmos. Environ., 226, 117393,
https://doi.org/10.1016/j.atmosenv.2020.117393, 2020.
Zdanovskii, A.: NOVYI METOD RASCHETA RASTVORIMOSTEI ELEKTROLITOV V
MNOGOKOMPONENTNYKH SISTEMAKH. 1, Zhurnal Fizicheskoi Khimii, 22, 1478-1485,

1948.

Zhang, Q., Canagaratna, M. R., Jayne, J. T., Worsnop, D. R., and Jimenez, J. L.: Time-and size-
resolved chemical composition of submicron particles in Pittsburgh: Implications for aerosol
sources and processes, J. Geophys. Res. Atmos., 1984–2012, 110, 2005a.
Zhang, Q., Worsnop, D. R., Canagaratna, M. R., and Jimenez, J. L.: Hydrocarbon-like and
oxygenated organic aerosols in Pittsburgh: insights into sources and processes of organic aerosols,
Atmos. Chem. Phys., 5, 3289-3311, 10.5194/acp-5-3289-2005, 2005b.
Zhang, Y., Tao, J., Ma, N., Kuang, Y., Wang, Z., Cheng, P., Xu, W., Yang, W., Zhang, S., Xiong,
C., Dong, W., Xie, L., Sun, Y., Fu, P., Zhou, G., Cheng, Y., and Su, H.: Predicting cloud condensation
nuclei number concentration based on conventional measurements of aerosol properties in the North
China Plain, Sci. Tot. Environ., 719, 137473, https://doi.org/10.1016/j.scitotenv.2020.137473, 2020.
Zhao, D. F., Buchholz, A., Kortner, B., Schlag, P., Rubach, F., Kiendler-Scharr, A., Tillmann,
R., Wahner, A., Flores, J. M., Rudich, Y., Watne, Å. K., Hallquist, M., Wildt, J., and Mentel, T. F.:
Size-dependent hygroscopicity parameter (κ) and chemical composition of secondary organic cloud
condensation nuclei, Geophys. Res. Lett., 42, 10,920-910,928,
https://doi.org/10.1002/2015GL066497, 2015.
Zhao, D. F., Buchholz, A., Kortner, B., Schlag, P., Rubach, F., Fuchs, H., Kiendler-Scharr, A.,
Tillmann, R., Wahner, A., Watne, Å. K., Hallquist, M., Flores, J. M., Rudich, Y., Kristensen, K.,
Hansen, A. M. K., Glasius, M., Kourtchev, I., Kalberer, M., and Mentel, T. F.: Cloud condensation
nuclei activity, droplet growth kinetics, and hygroscopicity of biogenic and anthropogenic



secondary organic aerosol (SOA), Atmos. Chem. Phys., 16, 1105-1121, 10.5194/acp-16-1105-2016,

2016.




Table 1. The density and the κ value of the related species used in this study.

| Species | Density (kg m$^{-3}$) | κ |
|---|---|---|
| $NH_4NO_3$ | 1720[a] | 0.58[b] |
| $NH_4HSO_4$ | 1780[a] | 0.56[b] |
| $H_2SO_4$ | 1830[a] | 0.90[b] |
| $(NH_4)_2SO_4$ | 1769[a] | 0.48[b] |
| Organics | 1400[a] | 0.10[b] |
| BC | 1770[c] | 0[d] |

[a] From Gysel et al. (2007); [b] From (Cai et al., 2018); [c] From Deng et al. (2019); [d] Assumed to be 0.





Table 2. Thermophysical properties of each component used in the multi-component evaporation
dynamics model.

| Parameters | ELVOA | LVOA | SVOA | Ammonium Nitrate | Ammonium Sulfate | Black Carbon |
|---|---|---|---|---|---|---|
| $C_i^*(T_{ref})$ (μg m$^{-3}$) [a] | $10^{-5}$ | $10^{-2}$ | 10 | 76 | $2\times10^{-3}$ | $10^{-30}$ |
| $D_i$ (m$^2$ s$^{-1}$)[b] | $5\times10^{-6}$ | $5\times10^{-6}$ | $5\times10^{-6}$ | $5\times10^{-6}$ | $5\times10^{-6}$ | $5\times10^{-6}$ |
| $\sigma_{s/a}$ (N m$^{-1}$)[c] | 0.05 | 0.05 | 0.05 | 0.05 | 0.05 | 0.05 |
| $M_i$ (kg mol$^{-1}$) | 0.2 | 0.2 | 0.2 | 0.08 | 0.132 | 0.28 |
| $\rho_i$ (kg m$^{-3}$) | 1400 | 1400 | 1400 | 1720 | 1769 | 1770 |
| $\Delta H_{vap,i}$ (kJ mol$^{-1}$)[d] | 80 | 80 | 80 | 152 | 94 | 100 |
| $\alpha$[e] | 0.09 | 0.09 | 0.09 | 0.09 | 0.09 | 0.09 |

[a] From Hong et al. (2017); [b] From Riipinen et al. (2010); [c] From Riipinen et al. (2010); [d] The $\Delta H_{vap}$
values of organics are obtained from the sensitivity test shown in Fig. S4 and the values of inorganic
species are from Hong et al. (2017); [e] Obtained from the sensitivity test shown in Fig. S4.





Table 3. The average and standard deviation values (mean±std) of $N_{CCN}$, AR, $D_{50}$, and $\kappa_{CCN}$ at 0.1%,

0.2%, 0.4%, 0.7%, 0.9% and 1.0% SS during the campaign.

| SS | 0.1% | 0.2% | 0.4% | 0.7% | 0.9% | 1.0% |
|---|---|---|---|---|---|---|
| $N_{CCN}$ (# cm$^{-3}$) | 2507±1187 | 4322±1981 | 5843±2461 | 6834±2921 | 7497±3210 | 7862±3352 |
| AR | 0.20±0.09 | 0.34±0.13 | 0.45±0.16 | 0.52±0.17 | 0.57±0.17 | 0.60±0.17 |
| $D_{50}$ (nm) | 145.55±11.26 | 92.83±8.80 | 66.79±6.33 | 52.56±5.46 | 45.38±4.82 | 42.26±4.45 |
| $\kappa_{CCN}$ | 0.48±0.13 | 0.47±0.15 | 0.31±0.10 | 0.22±0.09 | 0.20±0.08 | 0.20±0.08 |






FIGURE CAPTIONS
Figure 1. The temporal profile of the measured variables during the campaign. (a) particle number
size distribution; (b) $PM_1$ chemical composition measured by the SP-AMS along with mass
concentration of $PM_{2.5}$; (c) mass fraction of each species; (d) wind speed and direction. The color
code in (d) represents the wind direction.
Figure 2. The temporal profile of GF-PDF at the measured diameters (30, 50, 80, 100, 150 and 200
nm). The color code denotes the probability density and the red solid line represents the mean GF
($GF_{mean}$).
Figure 3. The average mass fraction distribution of SVOA, LVOA and ELVOA at the measured
diameters (30, 50, 80, 100, 150 and 200 nm), and average size-resolved hygroscopicity of organic
aerosol ($\kappa_{OA}$) with the upper and lower error bars (in red).
Figure 4. The campaign average diurnal variation of mass fraction of organics and f44 in bulk
$PM_1$ (a), the $\kappa$ values at 200 nm obtained by HTDMA ($\kappa_{HTDMA}$) and AMS ($\kappa_{AMS}$) measurements
(b), the PNSD (c) and mass distribution of organics (d). The shaded area represents standard
deviation.
Figure 5. The diurnal variation (displayed in boxplot) mass concentration of the deconvolved OA
factors from PMF analysis of AMS data, including more oxygenated OA (MOOA), less
oxygenated OA (LOOA), aged biomass burning OA (aBBOA), hydrocarbon-like OA (HOA),
biomass burning OA (BBOA), and nighttime OA (night-OA).
Figure 6. The average (solid line) and standard deviation (shaded area) diurnal variation of $\kappa_{OA}$ at
different particle diameters.





Figure 7. The size-resolved volatility distribution during daytime (8:00-16:00 LT) and nighttime
(20:00 to 4:00 LT) based on the median time of each cycle owing to the limited time resolution.
Figure 8. The average diurnal variation of $\delta_{N_{CCN}}$ at 0.1%, 0.2%, 0.4% and 0.7% SS based on fixed
$\kappa_{OA}$ (a), SR $\kappa_{OA}$ (b) and SR diurnal $\kappa_{OA}$ (c).





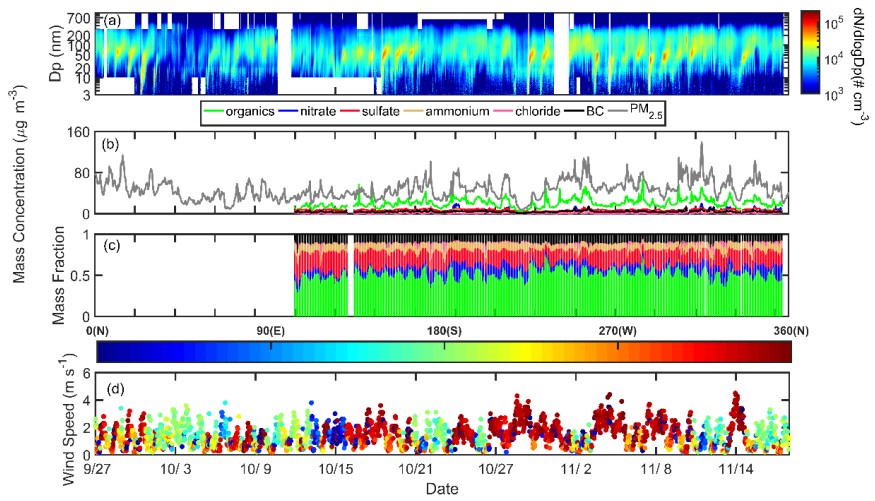




Fig. 1.





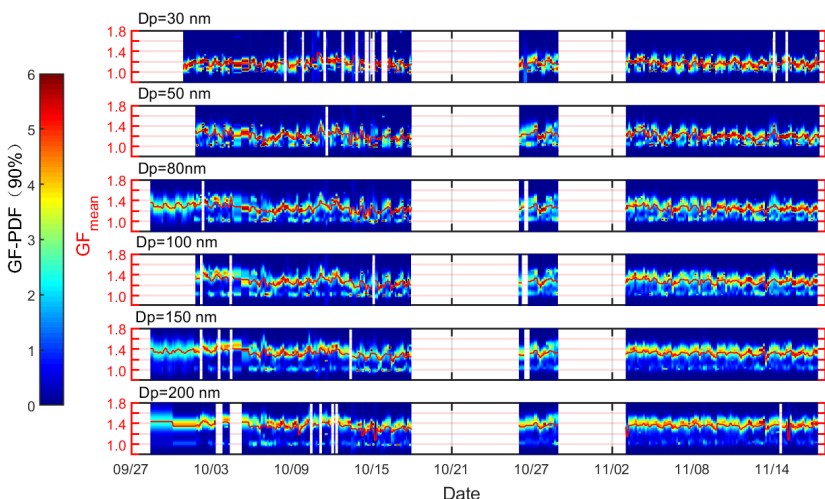


Fig. 2.





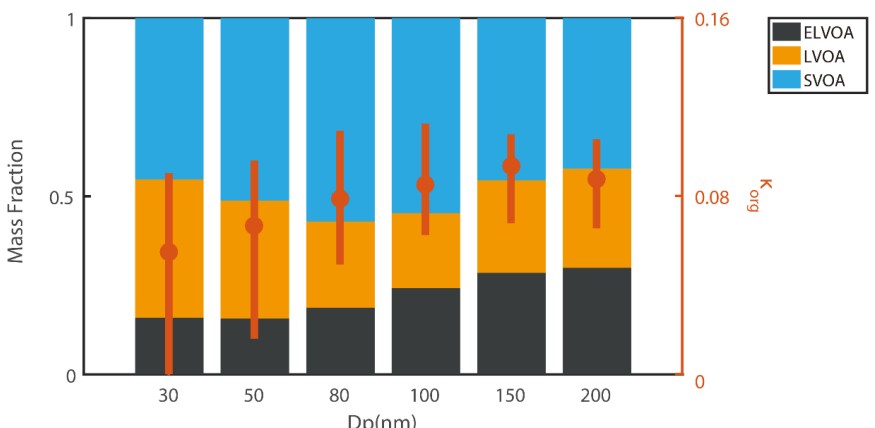



1047 Fig. 3.




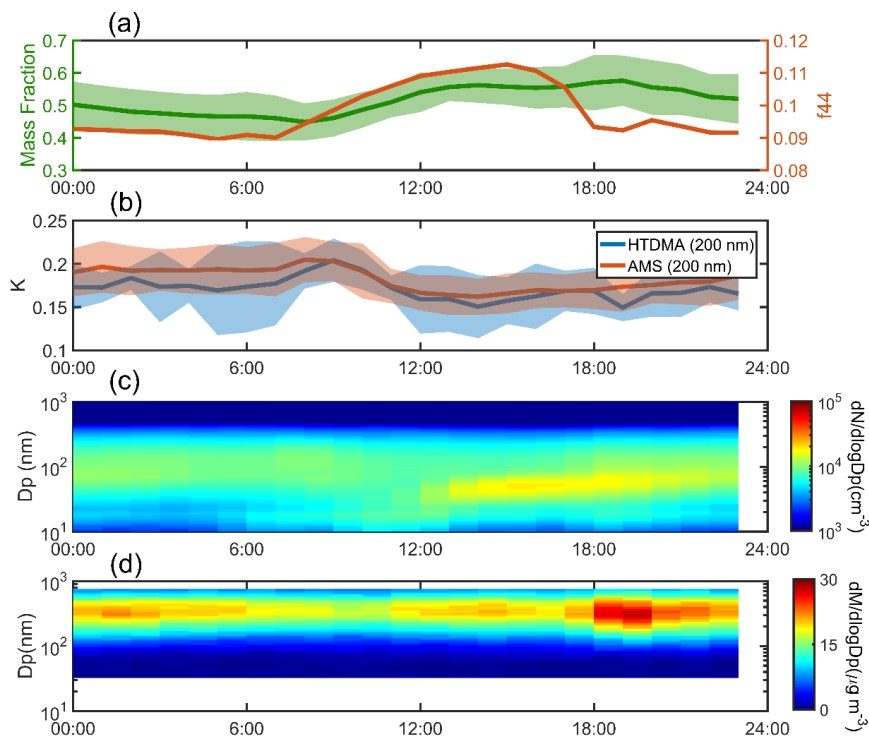



Fig. 4.



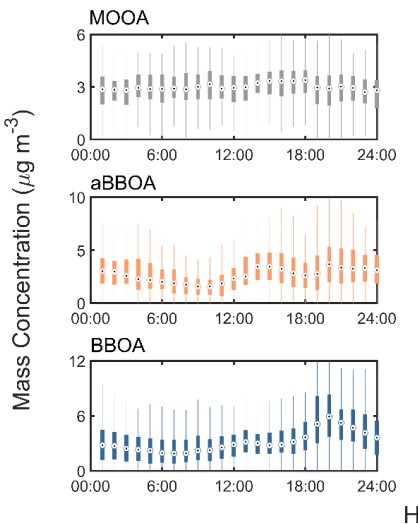

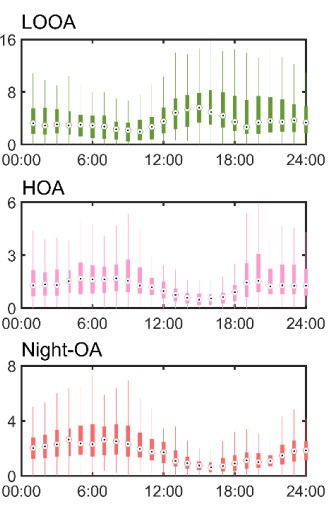



Fig. 5.





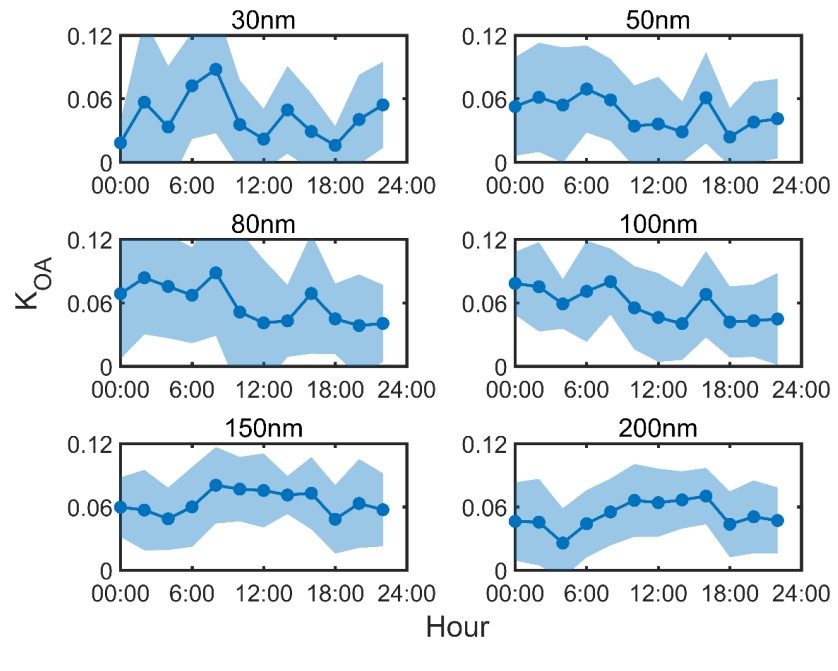



Fig. 6.





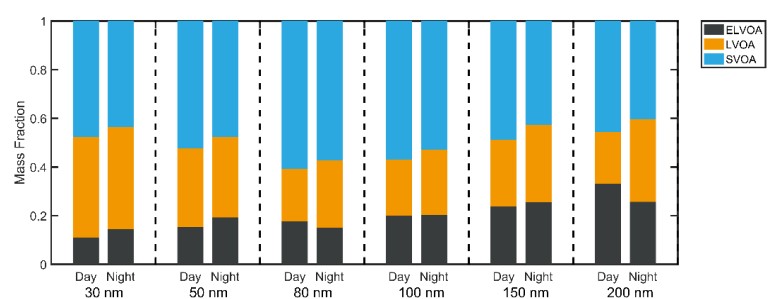



Fig. 7.






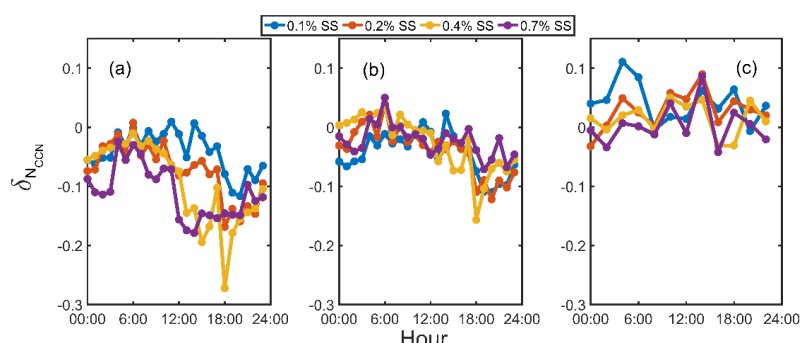



Fig. 8.