# Peer review of "Measurement Report: Distinct size dependence and"

_Atmospheric Chemistry and Physics, 2022_

## Referee Comment (RC2)

**Summary:**

This work demonstrates the hygroscopicity, volatility and CCN activity of OA particles at a rural environment of PRD. The manuscript fits well to the scope of ACP. However, I think more evidences or discussions should be included if possible. This paper is worth to be published, but not in its current form. Thus I recommend it to be accepted after the following comments listed below have been adequately addressed.

**Comments:**

- Section 2.2.2: Please give more information of reference data used in the Köhler theory when performing the CCNC calibration with ammonium sulfate particles. This is very important because different parameterizations will retrieve different critical supersaturations (Rose et al., 2008).
- 2. Line 340, I understand the decomposition of particles is hard to quantify. Could you roughly estimate the uncertainty?
- 3. Lines 409-413: I agree that the surfactant effect is crucial to explain the discrepancy in hygroscopicity closure study. Could you provide any chemistry evidence about that? Maybe from the AMS data or filter sample if this had been done in the campaign. Also, the difference between κCCN and κhtdma may also due to the parameterizations used in the CCNC and HTDMA calibration. See Wang et al., (2017). Please consider it and give more information as suggested in comment 1.

Many studies (Petters et al., 2009;Wex et al., 2009;Hersey et al., 2013;Wu et al., 2013;Hong et al., 2014;Hansen et al., 2015;Mikhailov et al., 2015;Pajunoja et al., 2015;Zhao et al., 2016) have reported the different hygroscopic properties from CCNC and HTDMA measurements. I would suggest more discussions should be added.

4. As shown in Fig. S10, the depression of surface tension is more obvious for larger particles (low SS). Is this more related to the ELVOCs? I would suggest to provide more case study, such as comparing the hygroscopicity (three methods) with different pollution condition (or OA content). For SS=0.7%, I do not suggest to adjust the

surface tension, it seems more reasonable to use the sigma of water.

5. Line 491: If there is a paper about the hygroscopicity of OA in the same study, please clarify the similarities and differences

Hansen, A. M. K., et al.: Hygroscopic properties and cloud condensation nuclei activation of limonenederived organosulfates and their mixtures with ammonium sulfate, Atmos Chem Phys, 15, 14071-14089, 10.5194/acp-15-14071-2015, 2015.

Hersey, Scott P., et al.: Composition and hygroscopicity of the Los Angeles Aerosol: CalNex, J Geophys Res-Atmos, 118, 3016-3036, 10.1002/jgrd.50307, 2013.

Hong, J., et al.: Hygroscopicity, CCN and volatility properties of submicron atmospheric aerosol in a boreal forest environment during the summer of 2010, Atmos Chem Phys, 14, 4733-4748, 10.5194/acp-14-4733-2014, 2014.

Mikhailov, E. F., et al.: Chemical composition, microstructure, and hygroscopic properties of aerosol particles at the Zotino Tall Tower Observatory (ZOTTO), Siberia, during a summer campaign, Atmos Chem Phys, 15, 8847-8869, 10.5194/acp-15-8847-2015, 2015.

Pajunoja, Aki, et al.: Adsorptive uptake of water by semisolid secondary organic aerosols, Geophys. Res. Lett., n/a-n/a, 10.1002/2015GL063142, 2015.

Petters, M. D., et al.: Towards closing the gap between hygroscopic growth and activation for secondary organic aerosol – Part 2: Theoretical approaches, Atmos Chem Phys, 9, 3999-4009, 10.5194/acp-9-3999-2009, 2009.

Rose, D., et al.: Calibration and measurement uncertainties of a continuous-flow cloud condensation nuclei counter (DMT-CCNC): CCN activation of ammonium sulfate and sodium chloride aerosol particles in theory and experiment, Atmos Chem Phys, 8, 1153-1179, 10.5194/acp-8-1153-2008, 2008.

Wang, Z., et al.: Dependence of the hygroscopicity parameter  $\kappa$  on particle size, humidity and solute concentration: implications for laboratory experiments, field measurements and model studies, Atmos. Chem. Phys. Discuss., 2017, 1-33, 10.5194/acp-2017-253, 2017.

Wex, H., et al.: Towards closing the gap between hygroscopic growth and activation for secondary organic aerosol: Part 1 – Evidence from measurements, Atmos Chem Phys, 9, 3987-3997, 10.5194/acp-9-3987-2009, 2009.

Wu, Z. J., et al.: Relating particle hygroscopicity and CCN activity to chemical composition during the HCCT-2010 field campaign, Atmos Chem Phys, 13, 7983-7996, 10.5194/acp-13-7983-2013, 2013.

Zhao, D. F., et al.: Cloud condensation nuclei activity, droplet growth kinetics, and hygroscopicity of biogenic and anthropogenic secondary organic aerosol (SOA), Atmos Chem Phys, 16, 1105-1121, 10.5194/acp-16-1105-2016, 2016.

---

## Author Comment (AC1)

Review of "Measurement Report: Distinct size dependence and diurnal variation of OA hygroscopicity, volatility, and CCN activity at a rural site in the Pearl River Delta (PRD) region, China" by Cai et al.

Cai et al. compile a report of findings of a comprehensive study of CCN activity for a measurement campaign in the Pearl River Delta (PRD) utilizing a complete suite of instruments to measure and understand the hygroscopicity and volatility from several differing methodologies. The work is of high quality and is complete and representative. I recommend it for publication with minor changes/corrections as listed below:

1. Line 82: "and plant" seems that some words are missing here, the intent of the sentence is unclear with the addition of plant.
Reply: We missed "wax" in this sentence. It has been revised to "… and plant wax…".

2. Lines 87-90: Here in the summary of several previous works it isn't mentioned where or under what conditions these varying hygroscopicities were reported.
Reply: We appreciate the reviewer for this valuable suggestion. These sentences in lines 87-91 have been revised to "Deng et al. (2018) reported a higher OA hygroscopicity ($\kappa_{OA}{\approx}0.22$) at about 150 nm than that ($\kappa_{OA}{\approx}0.19$) at sub-100 nm at a forest site. In contrast, Zhao et al. (2015) measured size-dependent hygroscopicity and chemical composition for SOA from various procedures and found that $\kappa_{OA}$ of SOA from α-pinene photooxidation decreased from 0.17 at 50 nm to 0.07 at 200 nm, which was attributed to the higher oxidation degree for smaller particles."

3. Line 174: I think that this is the line where PNSD needs to be defined, it is not defined anywhere in the paper.
Reply: It has been revised to "particle number size distribution (PNSD)".

4. Line 226: Why not report R to one more digit at least? (8.314)
Reply: It has been revised to "…gas constant (8.314 J mol$^{-1}$ K$^{-1}$)".

5. Line 228: meters not meter.
Reply: It has been revised to "meters".

6. Line 249: Here you state the assumption relative to internally mixed particles, but the GF HTDMA data indicates that at the very least some of the mixture was externally mixed. What effect if any does this have on the analysis?
Reply: We thank the reviewer for this suggestion. We predicted the $N_{CCN}$ using activation curve obtained by HTDMA measurement, which represented actual mixing states of the particles (Cai et al., 2018). We have modified the sentence in section 3.4 in line 578-579 and added a discussion in the supplement:
"The internal mixing assumption could slightly increase the predicted $N_{CCN}$ by about 6-10% (Sect. S3).

**Text in the supplement. Section S3 The impact of aerosol mixing state on the $N_{CCN}$ prediction**

The $N_{CCN}$ prediction is affected by the assumed particle mixing state (Wang et al., 2010). We estimated the impact of the mixing state assumption on the $N_{CCN}$ prediction by comparing the predicted $N_{CCN}$ based on AMS and HTDMA measurements. For the prediction based on AMS measurement, the particles were assumed to be internally mixed. In the latter approach, the mixing state was considered. The hygroscopicity parameter $\kappa_{critical}(Dp, SS)$ was defined as the point at which all particles could be activated at a specific diameter (Dp) and a specific SS. We calculated the $\kappa_{critical}(Dp, SS)$ using eq. (4) for a measured diameter (Dp) and a known SS. Particles with a $\kappa$ value higher than the $\kappa_{critical}(Dp, SS)$ were activated. The activation ratio ($AR_{HTDMA}(Dp, SS)$) for a known diameter and SS was obtained by integrating the $\kappa$-PDF for $\kappa > \kappa_{critical}(Dp, SS)$. Hence the predicted activation curve $AR_p(Dp, SS)$ was determined by fitting the $AR_{HTDMA}(Dp, SS)$ using eq. (6). Thus, the $N_{CCN}$ can be calculated:

$$N_{CCN,p}(SS) = \int_0^\infty AR_p(Dp_i, SS) n_i d\log Dp_i \tag{S3.1}$$

the detail of this approach could be found in Cai et al. (2018).

In general, the combination of the internal mixing assumption and fixed $\kappa_{OA}$ scheme would lead to an overestimation of $N_{CCN}$ (14%-23%, Fig. S3.1). Noting that adopting a fixed $\kappa_{OA}$ value could also overpredict $N_{CCN}$ (especially at high SS), which has been discussed in the text (section 3.4). This bias could be corrected by adopting SR $\kappa_{OA}$ scheme, which showed that the $N_{CCN}$ was overestimated by about 6%-10% (Fig. S3.1). Hence, we concluded that assuming the particle to be an internal mixture could lead to an overestimation of $N_{CCN}$ by about 6%-10%.

[Figure]

Figure S3.1. The predicted and measured $N_{CCN}$ at 0.1%, 0.2%, 0.4%, and 0.7% SS based on internal mixing assumption (blue and yellow dots) and actual mixing state (purple dots). The fixed $\kappa_{OA}$ scheme (blue dots) and SR $\kappa_{OA}$ scheme (yellow dots) were adopted in the prediction based on the internal mixing assumption."

7. Line 368: Hong Kong (two words).
Reply: It has been revised.

8. Line 376: Replace "It" with "This".
Reply: It has been revised.

9. Line 411: replace organics with organic.
Reply: It has been revised.

10. Line 418: This paragraph appears to make an argument that CCN measurements report kappa values that would be considered incorrect. If this is used to measure actual CCN activity and predict them, wouldn't a CCN instrument be the more appropriate measurement instead of an HTDMA, where the hygroscopicity of the HTDMA instrument would be the biased one? Understanding that

the kappa values are different, which is more appropriate for estimating modeled Nccn? (this is again brought up on line 560).

Reply: We appreciate the reviewer for this suggestion. The $\kappa$ values can be obtained at sub- and super-saturated conditions. The discrepancy between $\kappa_{CCN}$ and $\kappa_{HTDMA}$ does not suggest that the $\kappa_{CCN}$ or $\kappa_{HTDMA}$ is incorrect. It rather implies that water uptake ability of particles could be different under sub- and super-saturated conditions. Noting that the discrepancy between $\kappa_{HTDMA}$ and $\kappa_{CCN}$ could be caused by many factors, including the surfactant effect, parameterizations used in the CCNc and HTDMA calibration, the solubility of organics, and liquid-liquid phase separation (Liu et al., 2018;Petters and Kreidenweis, 2013;Rose et al., 2008;Pajunoja et al., 2015;Renbaum-Wolff et al., 2016).

The CCNc measurement is the most accurate method in the CCN activity measurement and prediction. However, the long-term CCNc measurement is expensive and requires human effort. The availability of CCNc measurement in field campaigns is still limited. Alternatively, the combination of PNSD measurement and chemical composition or hygroscopicity measurement can provide the estimation of $N_{CCN}$. On the other hand, the estimation of $N_{CCN}$ in the model is usually based on particle size distribution, composition, and supersaturation (Luo and Yu, 2011; Yu and Luo, 2009; Rastak et al., 2017; Abdul-Razzak and Ghan, 2002). The hygroscopicity of particles is calculated based on their chemical composition under sub-saturated conditions. However, the surfactant effect was found to increase the CCN activity relative to predictions derived for subsaturated condition, which would lead to uncertainty in the $N_{CCN}$ and climate simulation. Using different methods in predicting $N_{CCN}$ will help us to investigate this water uptake mechanism and improve the prediction of aerosol-cloud-climate interactions.

In order to avoid confusion, we have modified the sentences in lines 415-439,

"This significant discrepancy between the measured $\kappa_{CCN}$ and $\kappa_{HTDMA}$ values might suggest that the water uptake behavior is different under super- and sub-saturation conditions, which is likely attributed to the surfactant effect. It was reported that organic matter in the particles could serve as surfactant and lower surface tension by about 0.01-0.032 N m$^{-1}$, leading to a higher CCN activity and thus a higher $\kappa_{CCN}$ (Petters and Kreidenweis, 2013; Ovadnevaite et al., 2017; Liu et al., 2018). According to Eqs. (4) and (5), the $\kappa_{CCN}$ was more susceptibly affected by the value of surface tension than that of $\kappa_{HTDMA}$, which would lead to the discrepancy between $\kappa_{CCN}$ and $\kappa_{HTDMA}$ values. The surfactant effect is closely related to the presence of liquid-liquid phase separation (LLPS) for organic-containing particles at high RH (Renbaum-Wolff et al., 2016; Ruehl and Wilson, 2014). Once LLPS occurred, the organic film on the droplet surface would decrease surface tension and enhance water uptake. For particles of organic/inorganic mixture, the LLPS can occur when the O:C is lower than 0.8 (Bertram et al., 2011; Song et al., 2012a, b; Schill and Tolbert, 2013). The average O:C obtained using AMS is about 0.53 in this campaign, suggesting that the LLPS likely occurred at supersaturation conditions. Meanwhile, the variation of the discrepancy between $\kappa_{CCN}$ and $\kappa_{HTDMA}$ is statistically insignificant during clean and polluted periods (Fig. S7b and S7c), implying that the surfactant effect was hardly affected by pollution condition. Note that surface tension effect is not the only factor which leads to a higher $\kappa_{CCN}$. It was found that $\kappa_{CCN}$ could be higher than $\kappa_{HTDMA}$, since the existence of the slightly soluble compounds inhibits water uptake under subsaturation conditions (Zhao et al., 2016; Pajunoja et al., 2015; Dusek et al., 2011; Petters et al., 2009; Hong et al., 2014; Hansen et al., 2015). Other factors, such as different parameters used in the CCNc and HTMDA calibration and function groups associated with the carbon chain, can lead to a gap between

$\kappa_{HTDMA}$ and $\kappa_{CCN}$ (Rose et al., 2008; Wex et al., 2009). More future work is needed to better understand this water uptake mechanism and to improve the prediction of aerosol-cloud-climate interactions."

11. Line 602/603: This sentence needs to be reworked, it is unclear which statements and values correspond.
Reply: It has been revised to "For Aitken mode particles (30-100 nm), the $\kappa_{OA}$ values reached minimal (0.02-0.07) during daytime. Meanwhile, a daytime peak was observed for the $\kappa_{OA}$ value (~0.09) in the accumulation mode (150 and 200 nm), suggesting that the aging processes of preexisting particles were more dominant at accumulation mode particles."

12. Figure 4: In panel, A consider different colors (red/green color-blind issues).
Reply: We have revised the figure to improve the contrast as follows.

[Figure]

Figure 4. The campaign average diurnal variation of mass fraction of organics and f44 in bulk PM$_1$ (a), the $\kappa$ values at 200 nm obtained by HTDMA ($\kappa_{HTDMA}$) and AMS ($\kappa_{AMS}$) measurements (b), the PNSD (c) and mass distribution of organics (d). The shaded area represents standard deviation.

Reference:
    Abdul-Razzak, H. and Ghan, S. J.: A parameterization of aerosol activation 3. Sectional

representation, Journal of Geophysical Research: Atmospheres, 107, AAC 1-1-AAC 1-6, https://doi.org/10.1029/2001JD000483, 2002.

Cai, M., Tan, H., Chan, C. K., Qin, Y., Xu, H., Li, F., Schurman, M. I., Liu, L., and Zhao, J.: The size-resolved cloud condensation nuclei (CCN) activity and its prediction based on aerosol hygroscopicity and composition in the Pearl Delta River (PRD) region during wintertime 2014, Atmospheric Chemistry and Physics, 18, 16419-16437, 2018.

Liu, P., Song, M., Zhao, T., Gunthe, S. S., Ham, S., He, Y., Qin, Y. M., Gong, Z., Amorim, J. C., Bertram, A. K., and Martin, S. T.: Resolving the mechanisms of hygroscopic growth and cloud condensation nuclei activity for organic particulate matter, Nature Comm., 9, 4076, 10.1038/s41467-018-06622-2, 2018.

Luo, G. and Yu, F.: Simulation of particle formation and number concentration over the Eastern United States with the WRF-Chem + APM model, Atmos. Chem. Phys., 11, 11521-11533, 10.5194/acp-11-11521-2011, 2011.

Pajunoja, A., Lambe, A. T., Hakala, J., Rastak, N., Cummings, M. J., Brogan, J. F., Hao, L., Paramonov, M., Hong, J., and Prisle, N. L.: Adsorptive uptake of water by semisolid secondary organic aerosols, Geophys. Res. Lett., 42, 3063-3068, 2015.

Petters, M. D., and Kreidenweis, S. M.: A single parameter representation of hygroscopic growth and cloud condensation nucleus activity – Part 3: Including surfactant partitioning, Atmos. Chem. Phys., 13, 1081-1091, 10.5194/acp-13-1081-2013, 2013.

Rastak, N., Pajunoja, A., Acosta Navarro, J. C., Ma, J., Song, M., Partridge, D. G., Kirkevåg, A., Leong, Y., Hu, W. W., Taylor, N. F., Lambe, A., Cerully, K., Bougiatioti, A., Liu, P., Krejci, R., Petäjä, T., Percival, C., Davidovits, P., Worsnop, D. R., Ekman, A. M. L., Nenes, A., Martin, S., Jimenez, J. L., Collins, D. R., Topping, D. O., Bertram, A. K., Zuend, A., Virtanen, A., and Riipinen, I.: Microphysical explanation of the RH-dependent water affinity of biogenic organic aerosol and its importance for climate, Geophysical Research Letters, 44, 5167-5177, 10.1002/2017GL073056, 2017.

Renbaum-Wolff, L., Song, M., Marcolli, C., Zhang, Y., Liu, P. F., Grayson, J. W., Geiger, F. M., Martin, S. T., and Bertram, A. K.: Observations and implications of liquid–liquid phase separation at high relative humidities in secondary organic material produced by α-pinene ozonolysis without inorganic salts, Atmos. Chem. Phys., 16, 7969-7979, 10.5194/acp-16-7969-2016, 2016.

Rose, D., Gunthe, S., Mikhailov, E., Frank, G., Dusek, U., Andreae, M. O., and Pöschl, U.: Calibration and measurement uncertainties of a continuous-flow cloud condensation nuclei counter (DMT-CCNC): CCN activation of ammonium sulfate and sodium chloride aerosol particles in theory and experiment, Atmos. Chem. Phys., 8, 1153-1179, 2008.

Wang, J., Cubison, M., Aiken, A., Jimenez, J., and Collins, D.: The importance of aerosol mixing state and size-resolved composition on CCN concentration and the variation of the importance with atmospheric aging of aerosols, Atmospheric Chemistry and Physics, 10, 7267-7283, 2010.

Yu, F. and Luo, G.: Simulation of particle size distribution with a global aerosol model: contribution of nucleation to aerosol and CCN number concentrations, Atmos. Chem. Phys., 9, 7691-7710, 10.5194/acp-9-7691-2009, 2009.

---

## Author Comment (AC2)

*Summary:*

This work demonstrates the hygroscopicity, volatility and CCN activity of OA particles at a rural environment of PRD. The manuscript fits well to the scope of ACP. However, I think more evidences or discussions should be included if possible. This paper is worth to be published, but not in its current form. Thus I recommend it to be accepted after the following comments listed below have been adequately addressed.

*Comments:*

1. Section 2.2.2: Please give more information of reference data used in the Köhler theory when performing the CCNC calibration with ammonium sulfate particles. This is very important because different parameterizations will retrieve different critical supersaturations (Rose et al., 2008).

Reply: We appreciate the reviewer for this suggestion. We have modified the sentence in lines 179-181 and added a section in the supplementary about the calibration as follows:

"Before the measurement, the SMPSs were calibrated with PSLs (20, 50 and 200 nm) and the CCNc was calibrated with ammonium sulfate (($NH_4$)$_2SO_4$) particles at selected SSs (0.1%, 0.2%, 0.4%, 0.7%, 0.9%, and 1.0%, Sect. S1).

**Text in the supplement. Section S1 Supersaturation calibration of the CCNc**

Before and after the measurement, the CCNc was calibrated with ammonium sulfate (($NH_4$)$_2SO_4$) particles. The critical supersaturation ($Sc$) was calculated by Köhler theory:

$$Sc = exp\left[\left(\frac{4A^3}{27B}\right)^{1/2}\right], A = \frac{4\sigma_{s/a}M_w}{RT\rho_w}, B = \frac{6i_s n_s M_w}{\pi\rho_w} \tag{S1}$$

where $\sigma_{s/a}$ is the surface tension of the solution/air interface and is assumed to be pure water (0.0728 N m$^{-1}$ at 298.15 K) for simplicity, $M_w$ is the molecular weight of water (0.018 kg mol$^{-1}$), R is the universal gas constant (8.314 J mol$^{-1}$ K$^{-1}$), T is the thermodynamic temperature in Kelvin (298.15 K), $\rho_w$ is the density of water (about 997.04 kg m$^{-3}$ at 298.15 K), $i_s$ is the van't Hoff factor and is assumed to be 2.5, $n_s$ is the molality of ($NH_4$)$_2SO_4$, $n_s = \frac{\pi\rho_s D_{50}^3}{6M_s}$, $D_{50}$ is the critical diameter, $\rho_s$ is the density of ammonium sulfate (1769 kg m$^{-3}$), and $M_s$ is the mole mass of ammonium sulfate (0.132 kg mol$^{-1}$)."

2. Line 340, I understand the decomposition of particles is hard to quantify. Could you roughly estimate the uncertainty?

Reply: We appreciate the reviewer for this valuable suggestion. We have added some sentences in section 2.3.4 in lines 349-352 and a discussion in the supplement:

"We roughly estimated uncertainty caused by the decomposition and found that ignoring the decomposition of organics would lead to an underestimation of SVOA, while the decomposition of ($NH_4$)$_2SO_4$ played a minor role in the simulation (Sect. S2). However, the exact effects are still highly uncertain.

**Text in the supplement. Section S2 Estimation of the uncertainty caused by the decomposition**

During the heating process, ammonium sulfate (($NH_4$)$_2SO_4$) would decompose to ammonium bisulfate ($NH_4HSO_4$) or triammonium hydrogen sulfate (($NH_4$)$_3H(SO_4$)$_2$, and ammonia ($NH_3$). Meanwhile, extremely low volatile OA (ELVOA) would decompose into semi-volatile or low-volatile OA. This could lead to uncertainty in the simulation. To estimate the uncertainty, we simulate the campaign average data based on the following assumptions:

Case 1. All ($NH_4$)$_2SO_4$ would decompose to ($NH_4$)$_3H(SO_4$)$_2$ at 150°C and then sublimation, while the decomposition of organics is ignored.

Case 2. All ELVOA would decompose to SVOA at 100°C, while the decomposition of ammonium sulfate is ignored.

Case 3. All ($NH_4$)$_2SO_4$ would decompose to ($NH_4$)$_3H(SO_4$)$_2$ and sublimation at 150°C, and all of EVLOA would decompose to SVOA at 100°C.

The results show that the decomposition of ($NH_4$)$_2SO_4$ plays a minor role in the simulation if the decomposition of organics was ignored (Case 1). It is probably owing to the fact that ($NH_4$)$_2SO_4$ starts to volatilize at about 100°C and completely sublimate at about 200°C (Hong et al., 2017). The decomposition of organics would significantly increase the fraction of SVOA (Case 2 and 3) by about 0.15-0.54. However, the SSR increases from 0.0216 in the standard simulation (ignore decomposition) to 0.5277 and 0.6626 in the case 2 and 3, respectively, suggesting that the model fails to reproduce the MFR based on the adopted parameters ($\Delta H\_vap$=80 kJ mol-1 and α=0.09). Thus, the results in case 2 and 3 are highly uncertain. In short summary, the decomposition of ($NH_4$)$_2SO_4$ would lead to a minor uncertainty in the simulation, while the decomposition of organic matter would significantly affect the model results by increasing the fraction of SVOA, for which the exact effects were still unclear. Further investigations are needed to better understand the decomposition of particles during the heating processes.

[Figure]

Figure S2.1. The mass fraction distribution of SVOA, LVOA and ELVOA of the campaign averaged MFR based on different assumptions.
"

3. Lines 409-413: I agree that the surfactant effect is crucial to explain the discrepancy in hygroscopicity closure study. Could you provide any chemistry evidence about that? Maybe from the AMS data or filter sample if this had been done in the campaign. Also, the difference between $\kappa$CCN and $\kappa$htdma may also due to the parameterizations used in the CCNC and HTDMA calibration. See Wang et al., (2017). Please consider it and give more information as suggested in comment 1. Many studies (Petters et al., 2009;Wex et al., 2009;Hersey et al., 2013;Wu et al., 2013;Hong et al., 2014;Hansen et al., 2015;Mikhailov et al., 2015;Pajunoja et al., 2015;Zhao et al., 2016) have reported the different hygroscopic properties from CCNC and HTDMA measurements. I would suggest more discussions should be added.

Reply: We thank the reviewer for valuable suggestions. We totally agree that the discrepancy between $\kappa_{HTDMA}$ and $\kappa_{AMS}$ could be caused by many factors, including the surfactant effect, parameterizations used in the CCNc and HTDMA calibration, the solubility of organics, and liquid-liquid phase separation (Liu et al., 2018;Petters and Kreidenweis, 2013;Rose et al., 2008;Pajunoja et al., 2015;Renbaum-Wolff et al., 2016). The decrease of surface tension was positively correlated with the concentration of water-soluble organic aerosol (WSOA) (Facchini et al., 2000). Humic like

substances (HULIS) were found to be a major group of surface-active species (Sugo et al., 2019;Dinar et al., 2006). Unfortunately, identifying such surface-active species at the molecular level is still a challenge by using AMS data. We collected $PM_{2.5}$ filter samples in this campaign, while all the samples have been analyzed in other purpose. Thus, it is difficult to provide any direct chemical evidence about these surface-active species in this campaign. Alternatively, we analyzed the ratio of atomic oxygen to atomic carbon (O:C), which was associated with the occurrence of LLPS (Bertram et al., 2011). Once LLPS occurred, the organic film on the droplet surface would decrease surface tension and enhance water uptake (Liu et al., 2018). For particles of organic/inorganic mixtures, the LLPS can occur when the O:C is low than 0.8 (Bertram et al., 2011;Song et al., 2012a, b;Schill and Tolbert, 2013). During this campaign, the average O:C obtained by the AMS measurement is ~0.53, suggesting that the LLPS likely occurred at supersaturation conditions, which could lead the discrepancy between $\kappa_{HTDMA}$ and $\kappa_{CCN}$. We have modified some sentences and added some discussions in lines 415-439:

"This significant discrepancy between the measured $\kappa_{CCN}$ and $\kappa_{HTDMA}$ values might suggest that the water uptake behavior is different under super- and sub-saturation conditions, which is likely attributed to the surfactant effect. It was reported that organic matter in the particles could serve as surfactant and lower surface tension by about 0.01-0.032 N m$^{-1}$, leading to a higher CCN activity and thus a higher $\kappa_{CCN}$ (Petters and Kreidenweis, 2013; Ovadnevaite et al., 2017; Liu et al., 2018). According to Eqs. (4) and (5), the $\kappa_{CCN}$ was more susceptibly affected by the value of surface tension than that of $\kappa_{HTDMA}$, which would lead to the discrepancy between $\kappa_{CCN}$ and $\kappa_{HTDMA}$ values. The surfactant effect is closely related to the presence of liquid-liquid phase separation (LLPS) for organic-containing particles at high RH (Renbaum-Wolff et al., 2016; Ruehl and Wilson, 2014). Once LLPS occurred, the organic film on the droplet surface would decrease surface tension and enhance water uptake. For particles of organic/inorganic mixture, the LLPS can occur when the O:C is lower than 0.8 (Bertram et al., 2011; Song et al., 2012a, b; Schill and Tolbert, 2013). The average O:C obtained using AMS is about 0.53 in this campaign, suggesting that the LLPS likely occurred at supersaturation conditions. Meanwhile, the variation of the discrepancy between $\kappa_{CCN}$ and $\kappa_{HTDMA}$ is statistically insignificant during clean and polluted periods (Fig. S7b and S7c), implying that the surfactant effect was hardly affected by pollution condition. Note that surface tension effect is not the only factor which leads to a higher $\kappa_{CCN}$. It was found that $\kappa_{CCN}$ could be higher than $\kappa_{HTDMA}$, since the existence of the slightly soluble compounds inhibits water uptake under subsaturation conditions (Zhao et al., 2016; Pajunoja et al., 2015; Dusek et al., 2011; Petters et al., 2009; Hong et al., 2014; Hansen et al., 2015). Other factors, such as different parameters used in the CCNc and HTMDA calibration and function groups associated with the carbon chain, can lead to a gap between $\kappa_{HTDMA}$ and $\kappa_{CCN}$ (Rose et al., 2008; Wex et al., 2009). More future work is needed to better understand this water uptake mechanism and to improve the prediction of aerosol-cloud-climate interactions."

4. As shown in Fig. S10, the depression of surface tension is more obvious for larger particles (low SS). Is this more related to the ELVOCs? I would suggest to provide more case study, such as comparing the hygroscopicity (three methods) with different pollution condition (or OA content). For SS=0.7%, I do not suggest to adjust the surface tension, it seems more reasonable to use the sigma of water.

Reply: We thank the reviewer for this suggestion. The surfactant effect is associate with various factors, such as polarity of molecules, liquid density, vapor density, etc. (Egemen et al., 2000), while the volatility of OA is more related to functional groups, carbon number, oxidation state, and N:C ratio (Donahue et al., 2011;Donahue et al., 2012;Chuang and Donahue, 2016). Based on our current data, it is difficult to find direct evidence about the relationship between the depression of surface tension and the volatility of OA. We investigated the $\kappa$ values obtained by CCNc ($\kappa_{CCN}$), HTDMA ($\kappa_{HTDMA}$), and AMS ($\kappa_{AMS}$) during polluted (PM$_{2.5}$ > 60 $\mu$g m$^{-3}$) and clean periods (PM$_{2.5}$ < 30 $\mu$g m$^{-3}$). No significant variation of $\kappa_{CCN}$, $\kappa_{HTDMA}$, and $\kappa_{AMS}$ was observed between clean and pollute periods (Fig. S7), suggesting the surfactant effects might be hardly affected by pollution condition. For 0.7% SS, the overestimation of N$_{CCN}$ based on the reduced $\sigma_{s/a}$ is due to using a fix value of $\kappa_{OA}$ (0.1), while the $\kappa_{OA}$ is lower than 0.1 at corresponding size range. We predicted the N$_{CCN}$ at 0.7% SS by using the reduced $\sigma_{s/a}$ and SR $\kappa_{OA}$, and found that the overestimation could be corrected (Fig. S11).

We have added some sentences and modified the discussion in lines 409-418,

"The average $\kappa$ values obtained using HTDMA fall in a range of 0.1-0.17 at 30-200 nm (Fig. S7a), which were possibly attributed to a high fraction of organic matter (Fig. S6). The $\kappa_{AMS}$ is slightly higher than the $\kappa_{HTDMA}$ and the differences become larger with decreasing particle sizes. This was probably due to the overestimated $\kappa_{OA}$ at a small size range, which will be discussed in the next section. The hygroscopicity parameter $\kappa$ values obtained by the CCNc method were 0.48, 0.47, 0.31, 0.22, 0.20, and 0.20 at the above SS, respectively, which were much higher than those measured by the HTDMA in this study. This significant discrepancy between the measured $\kappa_{CCN}$ and $\kappa_{HTDMA}$ values might suggest that the water uptake behavior is different under super- and sub-saturation condition, which is likely attributed to the surfactant effect."

, and in lines 429-431,

"Meanwhile, the variation of the discrepancy between $\kappa_{CCN}$ and $\kappa_{HTDMA}$ is statistically insignificant during clean and polluted periods (Fig. S7b and S7c), implying that the surfactant effect was hardly affected by pollution condition.

[Figure]

Figure. S7 The mean and standard deviation values of $\kappa_{CCN}$, $\kappa_{HTDMA}$, and $\kappa_{AMS}$ during the campaign (a), clean (b) and pollute periods (c). The $\kappa$ values were pointed against their corresponding mean $D_{50}$ ($\kappa_{CCN}$) or selected diameter ($\kappa_{HTDMA}$ and $\kappa_{AMS}$). The dots represent the mean values, and the bars represent the one standard deviation. The relative clean and polluted periods were classified by the mass concentration of $PM_{2.5}$ ($< 30 \ \mu g \ m^{-3}$ and $> 60 \ \mu g \ m^{-3}$). "

, and in lines 588-590,

"The $N_{CCN}$ was slightly overestimated by using reduced $\sigma_{s/a}$ values, which was probably due to using a fixed $\kappa_{OA}$ values. This bias could be corrected by adopting SR $\kappa_{OA}$ scheme (Fig. S11).

[Figure]

Figure S11. The predicted and measured $N_{CCN}$ at 0.7% SS based on the $\sigma_{s/a}$ value (0.0728 N m$^{-1}$) for pure water and fixed $\kappa_{OA}$ (blue dots), reduced $\sigma_{s/a}$ value (0.059 N m$^{-1}$) and fixed $\kappa_{OA}$ (purple dots), and reduced $\sigma_{s/a}$ value (0.059 N m$^{-1}$) and SR $\kappa_{OA}$ (red dots).
"

5. Line 491: If there is a paper about the hygroscopicity of OA in the same study, please clarify the similarities and differences

Reply: We have modified the corresponding sentences in lines 512-515,

"For the same campaign, Kuang et al. (2021) reported the bulk $\kappa_{OA}$ of $PM_1$ based on aerosol optical hygroscopicity measurements, which could provide high time resolution data of $\kappa_{OA}$. The relationship between $\kappa_{OA}$ and different OA factors was investigated, which showed a negative correlation (R=-0.25) between LOOA and $\kappa_{OA}$, while a positive correlation (R=0.35) between aBBOA and $\kappa_{OA}$."

Reference:

Bertram, A. K., Martin, S. T., Hanna, S. J., Smith, M. L., Bodsworth, A., Chen, Q., Kuwata, M., Liu, A., You, Y., and Zorn, S. R.: Predicting the relative humidities of liquid-liquid phase separation, efflorescence, and deliquescence of mixed particles of ammonium sulfate, organic material, and water using the organic-to-sulfate mass ratio of the particle and the oxygen-to-carbon elemental ratio of the organic component, Atmos. Chem. Phys., 11, 10995-11006, 10.5194/acp-11-10995-2011, 2011.

Chuang, W. K., and Donahue, N. M.: A two-dimensional volatility basis set – Part 3: Prognostic modeling and NOx dependence, Atmos. Chem. Phys., 16, 123-134, 10.5194/acp-16-123-2016, 2016.

Dinar, E., Taraniuk, I., Graber, E. R., Katsman, S., Moise, T., Anttila, T., Mentel, T. F., and Rudich, Y.: Cloud Condensation Nuclei properties of model and atmospheric HULIS, Atmos. Chem. Phys., 6, 2465-2482, 10.5194/acp-6-2465-2006, 2006.

Donahue, N. M., Epstein, S. A., Pandis, S. N., and Robinson, A. L.: A two-dimensional volatility basis set: 1. organic-aerosol mixing thermodynamics, Atmos. Chem. Phys., 11, 3303-3318, 10.5194/acp-11-3303-2011, 2011.

Donahue, N. M., Kroll, J. H., Pandis, S. N., and Robinson, A. L.: A two-dimensional volatility basis set – Part 2: Diagnostics of organic-aerosol evolution, Atmos. Chem. Phys., 12, 615-634, 10.5194/acp-12-615-2012, 2012.

Egemen, E., Nirmalakhandan, N., and Trevizo, C.: Predicting Surface Tension of Liquid Organic Solvents, Environ. Sci. Technol., 34, 2596-2600, 10.1021/es991284u, 2000.

Facchini, M. C., Decesari, S., Mircea, M., Fuzzi, S., and Loglio, G.: Surface tension of atmospheric wet aerosol and cloud/fog droplets in relation to their organic carbon content and chemical composition, Atmos. Environ., 34, 4853-4857, 2000.

Liu, P., Song, M., Zhao, T., Gunthe, S. S., Ham, S., He, Y., Qin, Y. M., Gong, Z., Amorim, J. C., Bertram, A. K., and Martin, S. T.: Resolving the mechanisms of hygroscopic growth and cloud condensation nuclei activity for organic particulate matter, Nature Comm., 9, 4076, 10.1038/s41467-018-06622-2, 2018.

Pajunoja, A., Lambe, A. T., Hakala, J., Rastak, N., Cummings, M. J., Brogan, J. F., Hao, L., Paramonov, M., Hong, J., and Prisle, N. L.: Adsorptive uptake of water by semisolid secondary organic aerosols, Geophys. Res. Lett., 42, 3063-3068, 2015.

Petters, M. D., and Kreidenweis, S. M.: A single parameter representation of hygroscopic growth and cloud condensation nucleus activity – Part 3: Including surfactant partitioning, Atmos. Chem. Phys., 13, 1081-1091, 10.5194/acp-13-1081-2013, 2013.

Renbaum-Wolff, L., Song, M., Marcolli, C., Zhang, Y., Liu, P. F., Grayson, J. W., Geiger, F. M., Martin, S. T., and Bertram, A. K.: Observations and implications of liquid–liquid phase separation at high relative humidities in secondary organic material produced by α-pinene ozonolysis without inorganic salts, Atmos. Chem. Phys., 16, 7969-7979, 10.5194/acp-16-7969-2016, 2016.

Rose, D., Gunthe, S., Mikhailov, E., Frank, G., Dusek, U., Andreae, M. O., and Pöschl, U.: Calibration and measurement uncertainties of a continuous-flow cloud condensation nuclei counter (DMT-CCNC): CCN activation of ammonium sulfate and sodium chloride aerosol particles in theory and experiment, Atmos. Chem. Phys., 8, 1153-1179, 2008.

Schill, G. P., and Tolbert, M. A.: Heterogeneous ice nucleation on phase-separated organic-sulfate particles: effect of liquid vs. glassy coatings, Atmos. Chem. Phys., 13, 4681-4695, 10.5194/acp-13-4681-2013, 2013.

Song, M., Marcolli, C., Krieger, U. K., Zuend, A., and Peter, T.: Liquid-liquid phase separation and morphology of internally mixed dicarboxylic acids/ammonium sulfate/water particles, Atmos. Chem. Phys., 12, 2691-2712, 10.5194/acp-12-2691-2012, 2012a.

Song, M., Marcolli, C., Krieger, U. K., Zuend, A., and Peter, T.: Liquid-liquid phase separation in aerosol particles: Dependence on O:C, organic functionalities, and compositional complexity, Geophys. Res. Lett., 39, https://doi.org/10.1029/2012GL052807, 2012b.

Sugo, T., Okochi, H., Uchiyama, R., Yamanokoshi, E., Ogata, H., Katsumi, N., and Nakano, T.: The role of humic-like substances as atmospheric surfactants in the formation of summer-heavy rainfall in downtown Tokyo, City Environ. Interac., 3, 100022, https://doi.org/10.1016/j.cacint.2020.100022, 2019.